# Simulations of Leaf BSDF Effects on Lidar Waveforms

**Benjamin D. Roth** [1,*] **, Adam A. Goodenough** [1]**, Scott D. Brown** [1]**, Jan A. van Aardt** [1]**, M. Grady Saunders** [1] **and Keith Krause** [2]

[1]   Rochester Institute of Technology, Rochester, NY 14623, USA; aagpci@cis.rit.edu (A.A.G.); sdbpci@cis.rit.edu (S.D.B.); vanaardt@cis.rit.edu (J.A.v.A.); mgs8033@rit.edu (M.G.S.)
[2]   Battelle, NEON Program, Boulder, CO 80301, USA; kkrause@battelleecology.org
*   Correspondence: bdr3295@g.rit.edu

**Abstract:** Establishing linkages between light detection and ranging (lidar) data, produced from interrogating forest canopies, to the highly complex forest structures, composition, and traits that such forests contain, remains an extremely difficult problem. Radiative transfer models have been developed to help solve this problem and test new sensor platforms in a virtual environment. Many forest canopy studies include the major assumption of isotropic (Lambertian) reflecting and transmitting leaves or non-transmitting leaves. Here, we study when these assumptions may be valid and evaluate their associated impacts/effects on the lidar waveform, as well as its dependence on wavelength, lidar footprint, view angle, and leaf angle distribution (LAD), by using the Digital Imaging and Remote Sensing Image Generation (DIRSIG) remote sensing radiative transfer simulation model. The largest effects of Lambertian assumptions on the waveform are observed at visible wavelengths, small footprints, and oblique interrogation angles relative to the mean leaf angle. For example, a 77% increase in return signal was observed with a configuration of a 550 nm wavelength, 10 cm footprint, and 45° interrogation angle to planophile leaves. These effects are attributed to (i) the bidirectional scattering distribution function (BSDF) becoming almost purely specular in the visible, (ii) small footprints having fewer leaf angles to integrate over, and (iii) oblique angles causing diminished backscatter due to forward scattering. Non-transmitting leaf assumptions have the greatest error for large footprints at near-infrared (NIR) wavelengths. Regardless of leaf angle distribution, all simulations with non-transmitting leaves with a 5 m footprint and 1064 nm wavelength saw around a 15% reduction in return signal. We attribute the signal reduction to the increased multiscatter contribution for larger fields of view, and increased transmission at NIR wavelengths. Armed with the knowledge from this study, researchers will be able to select appropriate sensor configurations to account for or limit BSDF effects in forest lidar data.

**Keywords:** waveform lidar; lidar; radiative transfer model; forest remote sensing; leaf area index (LAI); bidirectional reflectance distribution function (BRDF); bidirectional scattering distribution function (BSDF); leaf optical properties

---

## 1. Introduction

### 1.1. Preface

As ecosystems worldwide become increasingly vulnerable due to threats such as global warming, increased human land use, pollution, invasive species, disease, parasitic insects, storm damage, and fire, monitoring and managing the Earth's forests arguably has never been more important. Forest management requires assessing growth conditions and taking proper action to ensure sustainable use of this natural resource. Accurately measuring forest health is a vital need, as resources are continually extracted, populations expand land use, and the impacts of climate change grow.

Many current assessment methods consist of ground campaigns to measure individual trees, which is time-consuming, costly, and often are a poor representation of the entire forest that needs to be characterized [1]. Emerging technologies in remote sensing and subsequent data processing have begun to fill this gap. Remote sensing is generally defined as detection, recognition, and evaluation at a distance [2]. Airborne and spaceborne remote sensing, in particular, have emerged as important modalities in measuring the condition of forests. These technologies contribute to assessments at multiple spatial scales, with the ability to measure variations over time. However, a focus on improving remote sensing technology must also include turning collected data into interpretive products for decision-making. Franklin [2] outlines some of the important aspects that remote sensing brings to the field of forest management. These include differentiating forest cover types and species; identifying locations of previous treatments of thinning, plantings, or cutovers; locating areas of insect damage, wind thrown, floods, and fire damage; and creating maps with metrics such as stand density, leaf area index (LAI), or biomass. As mentioned above, what is of particular use to forest scientists and practitioners is that remote sensing assessments can be performed at local, regional, and even continental scales.

Remote sensing via airborne and satellite platforms allows for this capability to make environmental assessments at the required scales. Relevant sensing systems and associated data processing algorithms continually are evolving to provide accurate and reliable information. One of the most important tools for advancing remote sensing systems are radiative transfer models (RTMs), which are able to simulate sensor, environmental, and scene characteristics. Numerous RTMs have been used in the development of sensors and data processing algorithms [3–12]. Radiative transfer models, specifically their use as a "virtual laboratory", have been used for improving a plethora of remote sensing applications, ranging from geophysics, urban development, and weather forecasting [9]. Traditional applications include simulating canopy reflectance under different sensor and illumination conditions [13], and the use of these models for plant growth predictions via examination of canopy development as a function of incident radiation [14]. RTMs also have been leveraged extensively for the development of sensors, e.g., for simulating the sensor response function of the LANDSAT TM sensor [15]. The use of these models extend past system simulations, however, where biophysical characteristics are often assessed via vegetation indices in RTMs, as well as the impact that environmental conditions, such as topographical changes, can have on such indices [16]. Directional scattering, furthermore, has been explored based on inversion parameters of forest scene characteristics, such as leaf density, angle distributions, and size [17]. More recent uses include synthetic image generation for the improved understanding and identification of constituents within chemical plumes via remote sensing technologies [18]. Another recent application is the creation of a lidar waveform simulator for the prediction of signals from NASA's Global Ecosystem Dynamics Investigation (GEDI) which contributed to data evaluation and algorithm development prior to launching the satellite [19].

However, simplifying assumptions are often used in simulations to enhance computational efficiency or because exact simulation properties, e.g., sensor, environmental, or scene-related, are not fully known. We contend that by evaluating current assumptions, specifically in terms of radiometric assumptions, an improvement in radiative transfer models can be realized, which will result in the development of better sensors and data processing algorithms.

*1.2. Errors Due to Leaf Scattering Assumptions in Optical Remote Sensing Simulations*

One application of radiative transfer models has been to study the link between remote sensing data and biophysical elements of forest canopies [4,20–24]. A major assumption in many of these models is perfectly diffuse scattering leaves. This assumption has been evaluated by Yang et al. [25] for medium resolution hyperspectral imagery. The authors evaluated the contribution of leaf specular reflection to the total canopy bidirectional reflectance factor (BRF). Imagery from the EO-1 Hyperion sensor and field measurements were used to identify parameters for the Stochastic Radiative Transfer Model (SRTM) [22,26]. Results showed an approximately 33% normalized root mean squared error

(NRMSE) at 550 nm and 660 nm, between a canopy with and without leaf specular components. However, polarization measurements are one possible solution to this challenge, since specular components are largely linearly polarized. Xie et al. [27] executed polarized BRF measurements on corn leaves, along with field measurements and photographic methods, to construct a nearly identical three-dimensional maize canopy. The radiosity–graphics combined model (RGM) [24] was used to perform sensitivity studies toward quantifying the difference between Lambertian leaf assumptions and leaf specular attributes. The study evaluated the dependence of the specular portion of the maize canopy BRF for different leaf angle distributions (LAD), leaf area index (LAI) values, leaf surface properties, and solar angles. LAD and specular component influences on canopy BRF previously has been investigated through Monte Carlo techniques, but efforts lacked computational power [28]. The authors discovered that near horizontal leaves, large solar zenith angles, and wavelengths in the visible spectral domain resulted in the largest contribution of specular reflectance [27]. Walter-Shea [29], in turn, measured corn and soybean leaf optical scattering properties, fit directional scattering properties with exponential curves, and then determined the contribution of the non-Lambertian component to the passive canopy optical reflected radiances, using the 1D radiative transfer model Cupid. The modeled results were compared to field measurements, and showed that the inclusion of the non-Lambertian leaves improved the model prediction, with up to a 7% difference. The author also assessed the contribution of leaf orientation, reporting up to a 20% effect at a nadir view, with horizontal leaves having the highest reflectance. *The errors identified in these studies demonstrate the compounding inaccuracy of biophysically derived values from remote sensing data when leaf specular contributions are ignored.* Although most vegetation canopy BRDF studies have concentrated on multispectral or hyperspectral remote sensing, lidar data also are not immune to specular influences.

### 1.3. Normalizing Lidar Intensity Data

Lidar has become popular in the remote sensing community as it adds the third spatial dimension to data. A few examples of airborne ground scanning systems operated by the US government include the NEON AOP (NSF) [30], EAARL (USGS) [31], LVIS (NASA) [32], G-LiHT (NASA) [33], and SLICER (NASA) [34], with several commercial companies now manufacturing sensors, including Leica, Optech, Riegl, and TopEye [35]. Lidar systems have even been placed in space with NASA's ICESAT [36], ICESAT-2 [37], and GEDI [38] missions. There are a variety of different lidar systems, often characterized by their laser wavelength, footprint size, pulse width, digitization method (discrete or waveform), and platform (terrestrial, airborne, and space based) [39]. However, intensity data from lidar are often not used, as these depend on a number of environmental and system conditions in addition to target material properties, e.g., adaptive gain settings [40]. Linking the intensity to material properties through normalization techniques therefore has been a focus of much research. For example, over large areas, landscape elevations can vary, thereby requiring a range normalization to lidar intensity data [41]. Besides range effects, incidence angle also presents problems when attempting to use lidar data for classification or segmentation. The effect of lidar incident angle to targets' BRDF effects previously has been investigated for extended targets that are larger than the beam footprint. These studies, some of which are discussed below, typically take advantage of waveform lidar's ability to capture the entire return energy from a target. However, many linear mode systems only record an arbitrary amplitude, which does not represent the true energy for different target geometries [42].

Jutzi and Gross [43] normalized first return intensity values from gabled roofs by considering range, incident angle, and atmospheric attenuation. The normalization was accomplished by considering the energy contained in Gaussian returns from a full-waveform system and then normalizing by range, incident angle, and atmosphere attenuation. The incident angle was found by determining the surface normal and known lidar scan direction (angle). If there are many points on a flat surface, the normal surface vector can be found as the last eigenvector of the covariance matrix of the three-dimensional points on the surface [44]. Both a Lambertian model and Phong specular–diffuse model were tested, but due to the diffuse nature of targets, the Lambertian correction was sufficient. Incident angle

normalization was also performed by Zhu et al. [45] in the interest of finding correlation between leaf water content and intensity. These experiments were done at close range in a laboratory environment, resulting in extremely high point densities on plant leaves. Planar surfaces were extracted to determine incidence angles. A diffuse–specular model, based on the Beckmann law [46], was created by interpolating reference spectralon panels at different angles and reflectances. The removal of the specular component allowed for an increase of correlation to leaf water content from an $R^2$ of 0.01 to 0.76. It thus follows that a similar normalization for airborne lidar systems (ALS), used for characterizing forest canopies, could prove invaluable for extracting data needed for forest management. However, the complexity of separating individual scattering areas, orientation, and reflectivity within an ALS footprint has yet to be solved. Instead, regression models to reference (ground) metrics are often used to extract canopy characteristics, such as leaf area index (LAI) and leaf area density (LADen).

### 1.4. Extracting LAI and LADen from Lidar Data

Lidar has been used to extract biophysical properties from forest canopies to include vegetation height [47], biomass [48], land cover classification [49], tree classification [50], and tree segmentation [51], to name a few. The leaf area index (LAI, one-sided leaf area per unit ground area) and leaf area density (LADen, one-sided leaf area per unit volume) [52] are forest metrics that can be estimated by lidar, but are susceptible to error due to leaf optical and physical properties. LAI and LADen are important parameters for ecological management and are often used as inputs to system models for understanding carbon sequestration and allocation processes [53,54]. LADen is especially important in understanding the vertical structure within forest canopies, holding the potential for more accurate above ground biomass estimates, as well as information on the impact of large-scale disturbances [55]. In fact, LADen can be mapped over the landscape in order to better understand the effects of forest disturbances (such as pathogens, invasive insects, fire, drought, and windthrow) and thus inform conservation decisions [56]. This may even be possible at a global scale, given the recent deployment of NASA's Global Ecosystem Dynamics Investigation (GEDI) system [38]. This bodes well for future forest assessments, since publicly available lidar data have greatly increased in the last few years, especially ALS data from which regional forest metrics can be derived.

Traditional methods of determining LADen involve lowering a plumb line and recording contact with vegetation or to collect photographs pointing upward in the canopy at different heights [57]. These methods are labor intensive, time-consuming, and can only be accomplished at fine scales. Remote sensing, specifically lidar, is able to overcome many of these problems by acting as a "plumb line" penetrating into the canopy, thereby producing information from which the three-dimensional internal structure can be interrogated [58]. Although lidar data holds great potential in standardizing and mapping LAI and LADen metrics, calibration of the data is still required with coincident ground reference data, often estimated from hemispherical photographs [59] or an LAI instrument such as Li-Cor [60]. These instruments approximate LAI from gap fractions, estimated by looking up into the canopy, and actually return a plant area index (PAI), which is often substituted for LAI. Furthermore, many considerations must be made when using these instruments, such as having a uniform sky, so that canopy gaps exhibit similar pixel values, while one also has to adjust for the lidar scan angle [61].

A repeatable and accurate method in estimating LADen from lidar point clouds was presented by Kamoske et al. by comparing LADen, derived from NEON AOP [30] and NASA G-LiHT [33] data, which differ in sensor specifications and canopy penetration [54]. They found a dependence on voxel resolution with higher correlations at coarser spatial scales, with an $R^2$ value of 0.9 at a 10 m horizontal spatial resolution. Such a result is possible because the lidar sensor and operating specifications are directly linked to penetration into the canopy. Larger scan angle diversity allows the lidar pulses to find gaps under the top canopy layer, and the greater the PRF, the more pulses will reach the forest floor. The most significant factor, however, is the beam spread combined with operating altitude, both of which directly affect the signal-to-noise ratio [62]. The NASA G-LiHT lidar system provides larger scan angles, higher PRF, and smaller beam spread than the NEON AOP. Though NASA G-LiHT

data seem superior for understanding internal forest structure, only point cloud data are available, while the NEON AOP provides small-footprint waveform data. Waveform data record the intensity of each return pulse with nanosecond resolution, thereby allowing for a more comprehensive look into the forest structure and complexities in finer temporal and spatial detail [63]. As a relatively new technology, the capability of (waveform lidar) wlidar data has yet to be fully exploited, and instead is usually down sampled to a discrete point cloud. We agree that a method for estimating LADen from the use of the entire lidar waveform intensities may show higher accuracy than estimates from point clouds alone [64].

*1.5. Simulating Lidar Signals*

Despite the increased use of lidar to extract meaningful structural assessments from forest canopies, there remains a gap in our ability to understand vegetation light interactions at fine scales. To this end, radiative transfer models have been created, which can help improve our understanding of signal interactions in the canopy, aid in the discovery of ideal sensor configurations, and contribute to the development of improved algorithms for extracting forest biophysical parameters. Simulations allow for the complete knowledge of scene geometry, as well as system parameters. Different modeling techniques include semiempirical, geometric, and Monte Carlo ray tracing (MCRT). Semiempirical and geometric models have broad assumptions and simplifications that do not lend themselves to small-scale interactions within the canopy. Semiempirical models consist of either Gaussian or lognormal signal profiles and produce lidar returns through a convolution between the lidar pulse and object distribution [65,66]. MCRT are more accurate, but require long rendering times. Some of the more prominent models include RAYTRAN [9], a MCRT model; FLiES [67], a 3D canopy MCRT with a coupled atmosphere model; RGM [68], a graphics based scattering model; GORT [69], a hybrid geometric optic and radiative transfer model that only considers first-order returns; DIRSIG [70], a photon mapping ray tracer model; POVRAY [71], a ray tracer model; LITE [72], a forward ray tracer with voxelized scattering probabilities; Librat [8], a MCRT designed to be flexible and modular; FLIGHT [73], a MCRT using voxelized properties and scene facets; and DART [4], a quasi-MCRT model.

The effects of environmental conditions, sensor configurations, and modeling methods on wlidar signals have previously been investigated with radiative transfer simulations. The following are a few examples of how simulations have increased our knowledge of lidar signals in forest canopies. Kotchenova et al. [74] demonstrated, through stochastic radiative transfer theory, the effects of multiscatter within the canopy on large-footprint wlidar signals. When multiscatter was applied, an increase in signal amplitude was observed, especially lower in the canopy. Calders et al. [75] found that crown archetypes and subsequent clumping are important factors in reproducing lidar signals through simulation. Disney et al. [8], on the other hand, used the Librat model to show the effect of signal triggering, scan angle, and footprint size on discrete lidar retrieval of canopy heights. Other authors, such as Qin et al. [76], evaluated the effect of scanning angle, flying altitude, and pulse density on canopy profile retrieval by modeling the scene and extracting waveforms with DART, while producing canopy profiles with GORT. Morsdorf et al. [77] used the POVRAY model to investigate the effect of footprint size on ground returns, reproducing the effect of increased ground returns with larger footprint sizes. Finally, Gastellu-Etchegorry et al. [39] compared DART large wlidar signals to actual waveforms from the Laser Vegetation Imaging Sensor (LVIS) and concluded that the inclusion of multiscatter in the model resulted in improved accuracy.

The Digital Imaging and Remote Sensing Image Generation (DIRSIG) model, developed by Rochester Institute of Technology, is especially suited for lidar phenomenology investigations as it seeks to simulate high-fidelity signals by incorporating all aspects of the imaging chain. As a synthetic imagery generation model, it is based on quantitative first principles, records sensor reaching radiance, captures material spectral characteristics, and accounts for directional reflectance properties. When simulating lidar signals, the model accounts for the geometrical form factor, multiscatter from physical surfaces, speckle properties from rough surfaces, atmospheric conditions, laser scintillation

and other beam effects, and allows for the inclusion of advanced sensor models [70]. DIRSIG supports a number of different lidar systems, including discrete, full waveform, and Geiger mode, with many past studies maturing and validating this capability [64,70,78–81]. Some of the previous wlidar studies with DIRSIG include an investigation by Wu et al. [82], who created simulated waveforms from which deconvolution techniques were evaluated. The authors also were able to extract branching structure and stem location estimates by preprocessing the waveforms through denoising, deconvolution, ground registration, and angular rectification. Romanczyk et al. [80] generated DIRSIG-simulated waveforms to study the effect of tree geometry components on the wlidar signal. They evaluated the complexity within scene models required to encompass the correct scale to maximize simulation efficiency, without sacrificing accuracy. Leaves were seen to be the dominant structure, while leaf stems, trunks, boughs, and first-order branching showed no statistical significance. Although the effect on wlidar signals from many types of environmental conditions has been examined, the impact of light directional scattering from leaves, known as the bidirectional scattering distribution function (BSDF), has been given little attention.

*1.6. Objectives*

Although BSDF has been shown to impact remote sensing imagery, as well as lidar intensity returns from targets filling the field of view (FOV) [42–45], little is known regarding how much this effect is present in airborne lidar systems (ALS) when interrogating forest canopies. There are also very few studies on transmission contributions to the return signal [72], and none that are known to the authors on specular transmission contributions. This may be true, since major limitations for forest canopy studies are their (forest) size and associated complications in obtaining reference data. Unlike Xie et al. [27], who were able to build a frame around a corn canopy to measure canopy BRF, these reference data are unavailable for forest canopies and lidar sensing. However, by implementing laboratory-measured leaf BSDF data onto facets in the DIRSIG model, we can perform sensitivity studies that reveal the impact of leaf specular components on lidar data. The objectives of this study therefore were to:

1.  Quantify intensity contribution from transmission, i.e., using opaque vs. realistic leaf transmissions with a waveform lidar sensor;
2.  Determine lidar waveform sensitivity for Lambertian vs. realistic BSDF leaves; and
3.  Analyze BSDF effects on LAI, derived from waveform intensity data.

Each of the objectives were evaluated by examining sensitivity to wavelength, lidar footprint, view angle, and leaf angle distribution (LAD). We hypothesized that (i) transmission has significant contributions to large-footprint lidar waveforms, but less so at small footprints, as photons scatter outside the sensor FOV, and (ii) lidar sensitivity to BSDF effects is greatest at visible wavelengths, small wlidar footprints, and oblique interrogation angles relative to the mean leaf angle. These assumptions were based on observations that (i) the BSDF becomes largely specular in the visible, (ii) small footprints have fewer leaf angles to integrate over, and (iii) oblique angles cause diminished backscatter due to forward scattering.

## 2. Materials and Methods

*2.1. Introduction to the Simulation Method*

Simulations with the DIRSIG model were performed for various sensor configurations and vegetation properties. A sensitivity study was performed to isolate the error found when purely diffuse scattering leaves, or non-transmissive leaves are assumed for waveform lidar simulations. Specifically, the effect on the waveform return signal and subsequent compounding error, when calculating LAI, were assessed. The dependence on wavelength, lidar footprint, view angle, and leaf angle distribution (LAD) all were explored. We used a 10 m thick uniform vegetation layer, raised 2 m off the ground

as the principal scene in these studies. The vegetation layer consists of 10 cm leaf disks of a spatially uniform distribution. We then analyzed leaf BSDF effects with a scene based on tree models in an effort to capture geometry that may be expected to occur naturally.

## 2.2. RAMI Comparison

Before completing the sensitivity study, we first did a cursory investigation into the legitimacy of lidar waveforms generated by DIRSIG. Ideally, simulations should be compared to lidar data from a real system. However, this would require exact knowledge of all physical structure, which is next to impossible when simulating forest environments. Past studies that have performed such comparisons required ad hoc normalizations, often approximating the scene with simple geometries [39]. Instead, we evaluated DIRSIG results in terms of models that participated in the radiation transfer model inter-comparison (RAMI-IV) campaign [83]. As part of RAMI-IV, lidar radiative transfer models were compared with a couple of abstract canopy scenes, one of which is named HET17, composed of overstory and understory vegetation representations over a uniform background. The scene was recreated for insertion into DIRSIG by modeling each specified facet. The overstory was created with spheres made up of uniformly distributed 10 cm diameter leaf disks, while the understory consisted of 1 m spheres, made from uniformly distributed 1 cm diameter disks. Both the overstory and understory had an LAI of 5. The abstract vegetation leaf disks had a near infrared (NIR) reflectance of 0.44, and transmittance of 0.5, while the ground had a reflectance of 0.16. For the inter-comparison, simulation configuration requirements call for a 50 m diameter instantaneous uniform cylinder beam and a 50 m diameter FOV, produced from a 2 m diameter detector and a 24 mrad FOV. These exact configurations are not possible within DIRSIG, since DIRSIG defines a laser beam from a source point with a divergence angle, and the FOV is defined by the detector and focal length. We approximated the required configuration with the parameters in Table 1. A number of simulations were run to find when the lidar signal reached an asymptote, based on the number of scattering events and photon bundles per pulse, along with the number of bounces each photon bundle could experience. We found for the scenarios we evaluated that 4,000,000 events, 8,000,000 photon bundles, and a maximum of 10 bounces were sufficient to produce high accuracy. These numbers are in line with those recommended by the DIRSIG lidar modality handbook [84]. We also had to add a manual search radius, from which density and vector directions are calculated, when interrogating the photon map, due to the nature of vegetation scenes containing hundreds of thousands of facets. We set this search radius to 1 cm, to ensure that it falls within the overstory leaf geometry, while also balancing run-time efficiency.

**Table 1.** Digital Imaging and Remote Sensing Image Generation (DIRSIG) lidar settings for radiation transfer model inter-comparison (RAMI-IV) comparison.

| Parameter | Value | Parameter | Value |
|---|---|---|---|
| Altitude | 2000 m | Gate Range | $1.323 \times 10^{-5}$ to $1.335 \times 10^{-5}$ s |
| Wavelength | 1064 nm | Bin Size | 4 ns (0.6 m) |
| Laser Spectral Width | 0.0003 μm | Receive Radius | 0.05 m |
| Pulse Energy | 100 mJ | Detector length | 100 μm square |
| Beam Shape | cylinder "Rect" | Focal Length | 0.004 m |
| Beam Divergence | 0.025 rad | Spatial subsampling | $100 \times 100$ |
| Pulse Length | $1 \times 10^{-21}$ s | Maximum events in photon map per pulse | 4,000,000 |
| Temporal Pulse Shape | Gaussian | Maximum source bundles per pulse | 8,000,000 |
| Photon Map (PM) search radius | 1 cm | Maximum bounces per photon bundle | 10 |

The simulation results are shown in Figure 1. In the multiscatter scenario, where light is able to transmit through leaves, more energy is preserved, thereby creating larger magnitudes. Also, notice a slight shift of the waveform down in height, due to the delayed response that results from multiple scattering. In the RAMI-IV study, lidar waveforms were normalized so that the integrated profiles equaled one. Different interpretations of the required configurations and reporting quantity led to the necessity for a normalization in which waveforms could be compared [83]. We converted our data in each bin to photons/meter, which we then normalized so that the integral of the waveform equaled one. The normalization causes a reduction in the DIRSIG multiscatter waveform as compared to the single scatter curve, mainly due to the near uniform increase in scale caused by the multiscatter return.

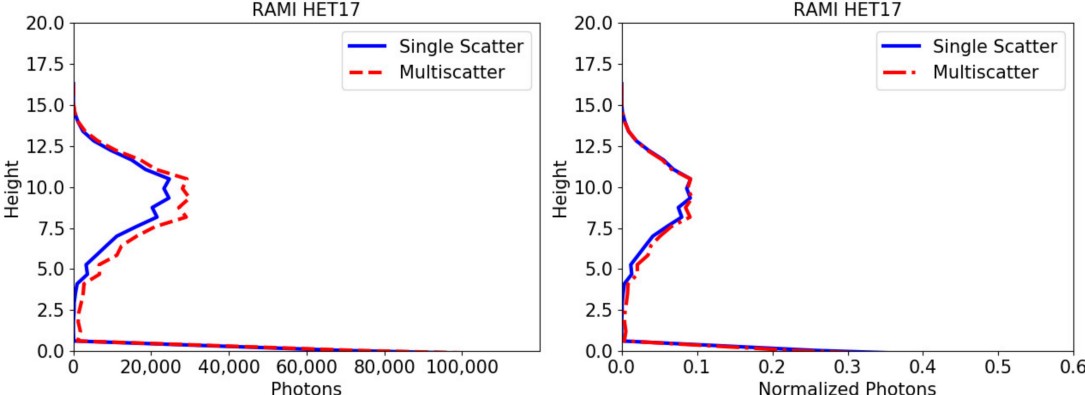

**Figure 1.** Lidar waveforms created by DIRSIG, approximating the configuration for the RAMI-IV study (both single and multiscatter) for the HET17 scene. The left plot is the waveform in terms of photons, while the right plot waveforms are normalized to have an integral of one, as was performed in RAMI-IV.

Overlaying the normalized DIRSIG waveforms onto the lidar plots from the RAMI-IV study, seen in Figure 2, shows that peak magnitudes in the canopy are similar, but that the DIRSIG returns appear to be lower in the canopy, with some noticeable structure. The structure seen in the DIRSIG waveform, as compared to the other models, is most likely caused by modeling the scene as distinct facets, instead of a turbid medium and possibly due to the coarse 0.6 m bins used, in an effort to record the canopy in 20 range bins per the RAMI IV requirements. The downward shift may be a time of flight recording difference, which is most noticeable when looking at the ground returns. The shift likewise may be attributed to different ground definitions, and subsequent alignment to the leading edge of the ground return for the original comparison, which we did not perform. In general, DIRSIG exhibits the expected phenomenology when adding multiscatter, and produces similar returns to those reported in the RAMI-IV study.

### 2.3. Multiscatter Contribution Comparison

Another validation was completed by simulating the multiscattering contribution for different beam footprints and leaf scattering albedo, which were then compared to a sensitivity study previously published using the DART model [39]. The study was accomplished by varying the lidar beam size (footprint radius) and the total albedo, defined as the sum of the reflectance and transmittance, with the reflectance and transmittance always being equal. The FOV was set to be always twice that of the beam footprint by varying the focal length. We simulated the platform at 1000 m, with a nadir-viewing wlidar system. The parameters set in DIRSIG are seen in Table 2. The scene consisted of a 10 m height vegetation layer, made up of 10 cm diameter leaf disks with an LAI of 4, and having spherical leaf angle distribution all over a perfectly absorbing ground. Because we simulated actual leaf facets, we found it overly computationally burdensome to create the vegetation layer beyond 30 × 30 m. The limit in scene size precluded modeling beyond a 7.5 m beam footprint radius, which was coupled with a 30 m



FOV. The multiscatter percent contribution was calculated by simulating both the single scatter and multiscatter waveforms, taking the sum of each waveform, finding the difference, and then dividing by the sum of the multiscatter waveform. The result is shown in Figure 3. Comparing the result to that found in the DART study shows that the DIRSIG multiscatter contribution is about 75% of what had been previously reported. We attributed the discrepancy to the differences between the scenes and the simulation settings. The DART study used a turbid medium for the vegetation layer, while we modeled individual facets. Moreover, the limited size of our scene may have reduced the multiscatter contribution at the large footprints. We also set a manual search radius of 1 cm, from which density and vector directions were calculated when interrogating the photon map. We were able to achieve slightly higher values by decreasing the search radius even further, but noise became an issue (e.g., a 0.5 cm search radius resulted in 38.5% maximum multiscatter contribution). Furthermore, the precise lidar configurations were not specified in the DART study, so we were unable to replicate the study exactly. However, the general shape in the trend of an increasing multiscatter contribution for increasing footprint radius and total albedo matched between the studies. We also display the one albedo, 7.5 radius footprint waveforms in Figure 3, thereby showing the expected increase in magnitude and downward shift when multiscatter is accounted for.

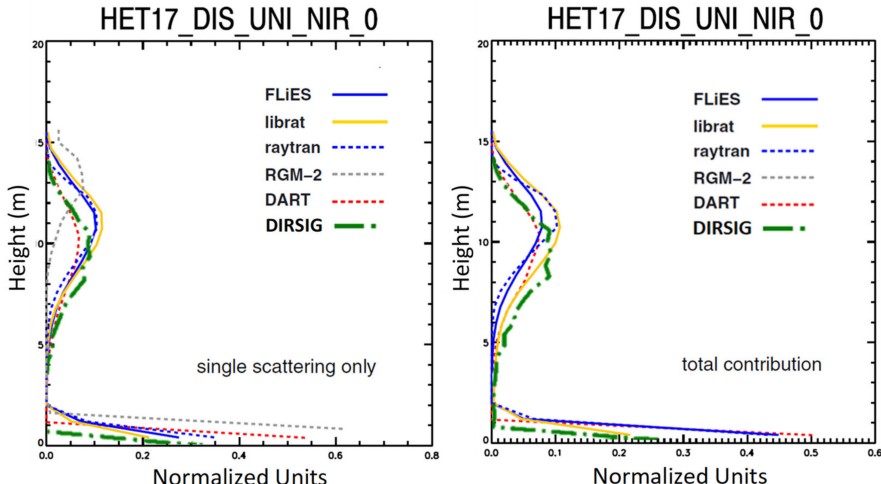

**Figure 2.** Normalized lidar waveforms from the RAMI-IV study with the DIRSIG-produced waveforms overlapped. The left plot displays waveforms produced from single scatter contributions, while the right figure includes multiscatter (figure adapted from [83]).

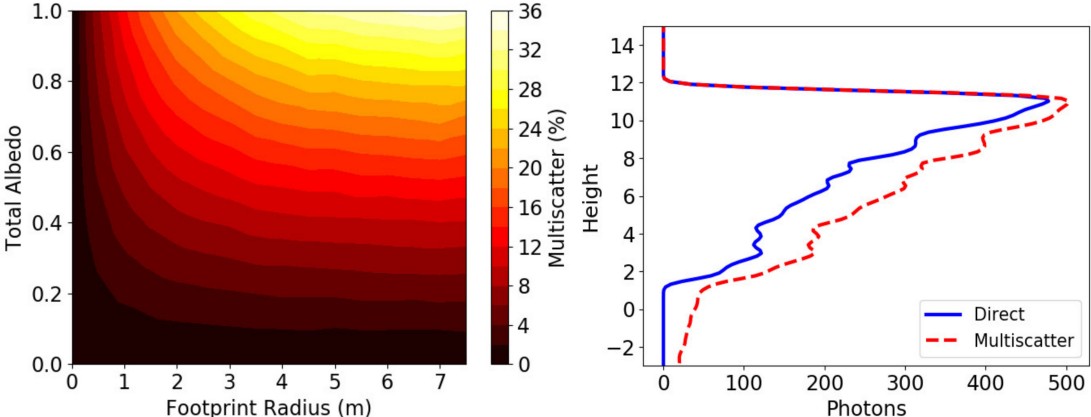

**Figure 3.** The left figure is the multiscatter contribution percentage as a function of lidar footprint radius and total albedo (reflectance + transmittance); the right figure is the waveform for the one albedo, 7.5 footprint radius.

**Table 2.** DIRSIG lidar settings for RAMI-IV comparison.

| Parameter | Value | Parameter | Value |
|---|---|---|---|
| Altitude | 1000 m | Gate Range | $6.55 \times 10^{-6}$ to $6.75 \times 10^{-5}$ s |
| Wavelength | 1064 nm | Bin Size | 1 ns |
| Laser Spectral Width | $1 \times 10^{-5}$ µm | Receive Radius | 0.025 m |
| Pulse Energy | 0.2 mJ | Detector length | 250 µm square |
| Beam Shape | Gaussian | Focal Length | varied |
| Beam Divergence | varied | Spatial subsampling | $101 \times 101$ |
| Pulse Length | 3 ns | Maximum events in photon map per pulse | 400,000 |
| Temporal Pulse Shape | Gaussian | Maximum source bundles per pulse | 800,000 |
| PM search radius | 1 cm | Maximum bounces per photon bundle | 10 (for multiscatter) |

### 2.4. Data-Driven BSDF in DIRSIG: Description and Verification

A recent addition to DIRSIG, specifically created for the DIRSIG5 release, is the ability to add data-driven BSDF descriptions to the material properties of facets within the scene. This allows for an open-ended insertion of optical scattering properties. Because DIRSIG5 has yet to support lidar, all simulations were completed in DIRSIG4. A beta version of the data-driven BSDF was incorporated into DIRSIG4, which we used to import our realistic leaf model into the simulations. The data-driven model uses a generic bidirectional weighting function, incorporating the projected area. The inputs include wavelengths, incident angles, view vectors, and associated BSDF data that are then processed into a spherical quad-tree (SQT) for efficient simulation processing [85]. The structure of the data is geometrically adaptable, thus concentrating tree nodes at high gradients. Also, for sampling efficiency, the file is set up with a hierarchical partitioning scheme to enable fast queries.

We tested the implementation of integrating measured BSDF data into DIRSIG with simulations of both a monostatic and bistatic lidar. We first used a monostatic lidar simulation to evaluate the sum of the return pulse when scanning along a single azimuth with different zenith angles. The system looked at a single plate many times larger than the lidar spot size of 0.5 m and a ground sampling distance (GSD) of 1 m, from a fixed distance of 500 m. The leaf BSDF material properties, measured from a sweetgum (*Liquidambar styraciflua*) leaf and fit to the Smith GGX model for BRDF and modified Dual Microfacet model for the bidirectional transmittance distribution function (BTDF), were applied to the plate [86,87]. Radiometric samples were taken in 5° zenith steps, thereby capturing the backscatter from 0° to 85°. The same simulation was completed, but with a change in the plate properties to that of a perfectly reflective and diffuse surface, from which the backscattered BRDF, also known as the monodirectional reflectance distribution function (MRDF) [88], could be determined. The resulting MRDF then was compared to the MRDF, taken directly from the BRDF input "RAW" file, which was used to assign the BSDF to the plate. Plots of the comparisons made at 550, 1064, and 1550 nm are shown Figure 4. Overall, the simulated MRDF values agree well with the input values from the RAW file, with slightly more noise at 1064 and 1550 nm, due to the SQT sampling, since the MRDF structure is a smaller percentage of the MRDF magnitude at these wavelengths.

We also verified the data-driven BSDF material properties by creating a "virtual" goniometer within DIRSIG, per a bistatic lidar configuration. The scene consisted of a perfectly absorbing ground and a flat plate at an altitude of 500 m, with both reflectance and transmittance defined with SQT material properties based on a modeled sweetgum leaf. The lidar source was fixed at a distance of 499 m from the center of the plate, and the receiver moved to various viewing azimuths and zenith angles,

also 499 m from the plate. By looking at the center of the plate, both above and below the plate, both reflectance and transmittance were able to be measured. The signal at each viewing location was taken as the sum of the lidar waveform. A BRDF was then extracted by running the same scenarios with a perfectly Lambertian reflecting material property assigned to the plate. The BRDF and BTDF from the raw file, compared to the resulting BRDF and BTDF from the simulated lidar goniometer for a 45° source and view angles in 30° azimuth steps (0°–360°) and 15° zenith steps (0°–60°), are shown in Figure 5 for 550 nm and Figure 6 for 1064 nm wavelengths. The figures display polar plots of the BRDF and BTDF at every viewing azimuth, 0°–360° clockwise starting at the top, and zenith, 0°–60° starting from the center. Spline interpolation is used in one degree azimuth and zenith increments to fill the simulated goniometer plots. Note the forward scattering of the light with the source and maximum signal at opposing azimuths.

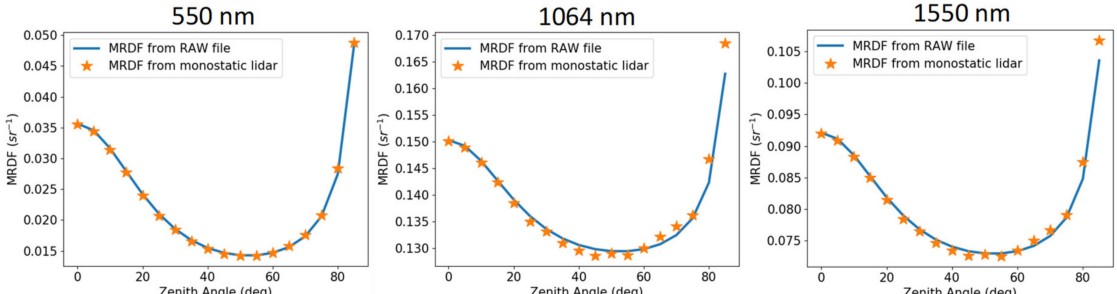

**Figure 4.** Comparison of monodirectional reflectance distribution function (MRDF) seen from a simulated lidar in DIRSIG, scanning at 5° zenith steps and capturing the backscatter from 0° to 85° zenith angles of a modeled flat plate. The resulting MRDF is compared to the input raw file, defining the bidirectional scattering distribution function (BSDF) properties of the plate, at equivalent data point locations. The comparison was accomplished at 500, 1064, and 1550 nm, as seen in the plots from left-to-right.

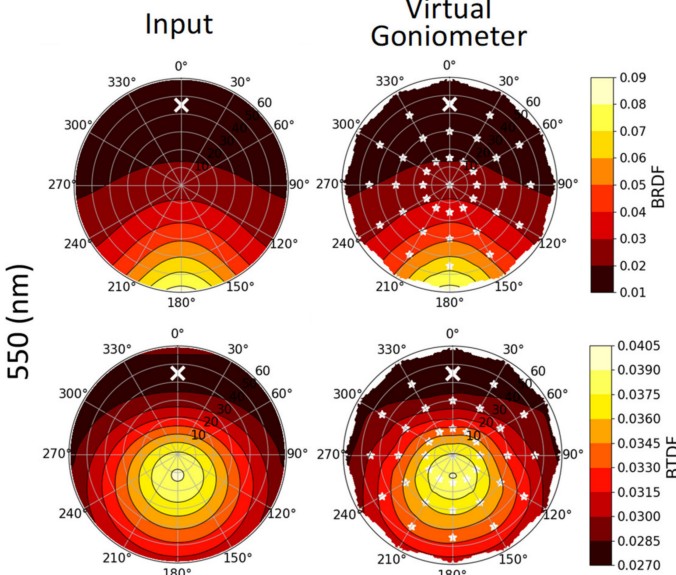

**Figure 5.** Comparison at 550 nm of bidirectional reflectance distribution function (BRDF), top row, and bidirectional transmittance distribution function (BTDF), bottom row, between input BSDF material properties, left column, and a simulated goniometer in DIRSIG, right column. The illumination source is set at 45° with view angles in 30° azimuth steps (0°–360°) and 15° zenith steps (0°–60°) for the goniometer. The white "x" depicts the source angle, while white stars represent the measured locations of the goniometer.

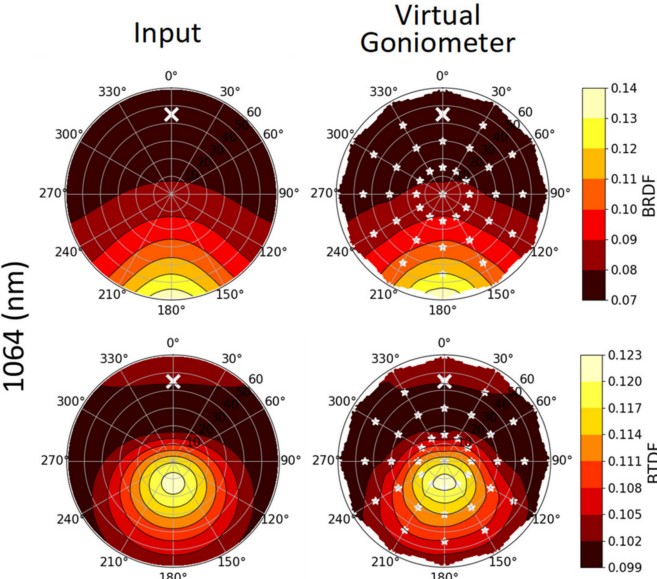

**Figure 6.** Comparison at 1064 nm of BRDF, top row, and BTDF, bottom row, between input BSDF material properties, left column, and a simulated goniometer in DIRSIG, right column. The illumination source is set at 45° with view angles in 30° azimuth steps (0°–360°) and 15° zenith steps (0°–60°) for the goniometer. The white "*x*" depicts the source angle, while white stars represent the measured locations of the goniometer.

## 2.5. Leaf Angle Distribution Usage

Part of this study evaluated the effect that LADs have in connection with leaf directional scattering properties on the returned wlidar signal. We define the distributions according to the "graphical" method used by Verhoef [89], who takes the basis as the cumulative distribution function (cdf) of the uniform distribution, and builds the trigonometric functions ($a \sin x$) and ($\frac{1}{2} b \sin 2x$) on top with a coordinate transformation. Then all distributions are defined by two variables, $a$ and $b$, from which practically all distributions can be defined. Verhoef [89] provides a simple algorithm which we implemented to produce the desired cdf, as seen in Figure 7. Of the leaf angle distributions first named by De Wit [90], spherical, planophile, and plagiophile distributions specifically were examined in this study, as most deciduous broadleaf trees can be described by one of these three distributions [91].

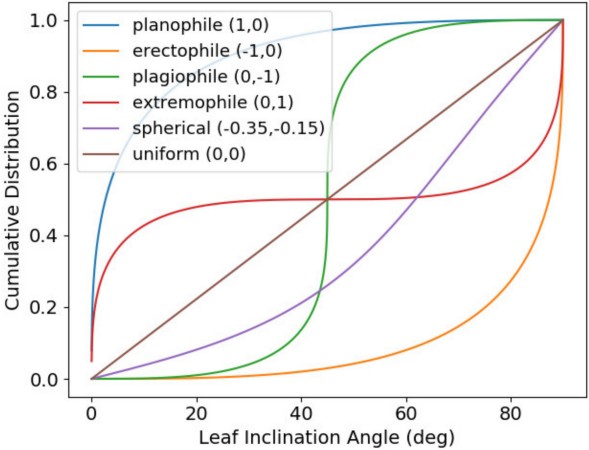

**Figure 7.** Leaf angle distribution (LAD) cdfs of some of the more prominent named distributions. Each distribution is defined here by its name, e.g., planophile, and its two variables (*a,b*) according to the method described by Verhoef [89].

### 2.6. Leaf Area Density and Leaf Area Index

One of the metrics used to evaluate lidar waveforms was an LAI derived directly from the waveform, given a reference. LAI and leaf area density (LADen), the "Den" in the acronym to distinguish from LAD, were calculated via a voxel-based Beer–Lambert law approach, closely resembling that first introduced by MacArthur and Horn [57], and also used in a number of other publications [54,61,92,93]. Richardson et al. [94] showed that the Beer–Lambert law approach, when compared to a number of other techniques to estimate LAI from airborne lidar, had the most correlation to LAI reference data from ground-based hemispherical photographs. We specifically used the formulism laid out by Kamoske et al. [54]:

$$LADen_{i-1} = \ln\left(\frac{S_e}{S_t}\right)\left(\frac{1}{k\Delta z}\right), \tag{1}$$

where $\Delta z$ is the voxel height and $k$ is an extinction coefficient estimated from the reference data. In the Kamoske et al. [54] study, $S_e$ and $S_t$ were the number of pulses entering and exiting a voxel, respectively, otherwise known as "hit counting". However, instead of hit counting, we propose a novel method to estimate LADen from a single waveform that takes advantage of wlidar's ability to record the entire pulse by using the intensity values. Then, $S_e$ and $S_t$ will be the integrated intensity entering and exiting a voxel, otherwise known as the cumulative power distribution after $R_1$, $CPD_{after}(R_1)$ and the cumulative power distribution after $R_2$, $CPD_{after}(R_2)$ [64]. The intensity, or cumulative power distributions entering and exiting a voxel, were calculated after the same manner in which Hagstrom [64] produced transmission voxels, as illustrated in Figure 8. A disadvantage of point counting is that it requires ground returns in the vertical column to ensure that the entire vertical structure was sampled, thus requiring a ground return. The requirement to have ground points in a vertical voxel column is " ... *the most significant limiting factor in the estimation of LADen ...* " [54]. Moreover, the thicker the canopy, the more error results from a point counting method. Intensity accounting, on the other hand, does not have this constraint to calculate LADen; however, ground returns are needed if an LAI is estimated for the entire column, and is needed for the reference waveforms to calculate an extinction coefficient. Intensity accounting therefore has potential to increase accuracy between sensors at much finer resolutions. Hagstrom [64] showed that using the CPD instead of hit counting causes the calculated transmission through a voxel to converge to the truth much quicker. Far fewer pulses are needed, thereby allowing for smaller voxels, and hence a higher resolution of 3D LADen. Once the voxelated LADen is calculated, the vertical summation yields LAI estimates.

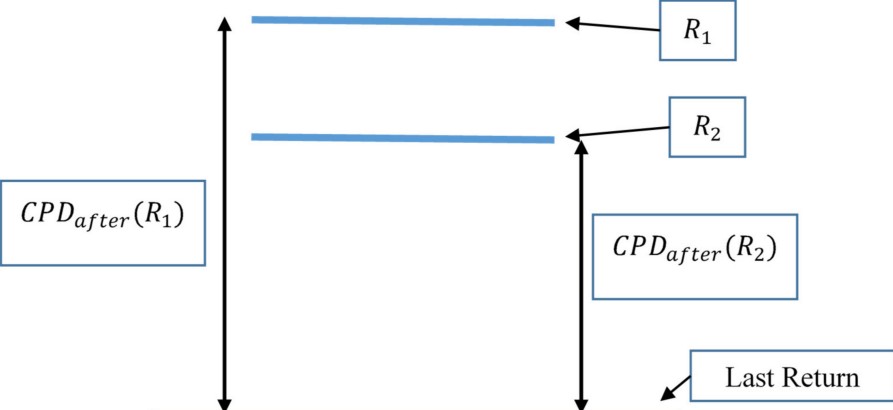

**Figure 8.** The $CPD_{after}(R_1)$ is the integrated waveform after the top of the voxel, while $CPD_{after}(R_2)$ is the integrated waveform after the bottom of the voxel.

The extinction coefficient, $k$, in Equation (1) is determined from ground reference data. This can be done with a single waveform, if a reference LAI is provided. The $k$ value is first set to one to find an uncalibrated LADen from which an uncalibrated LAI is determined. A top and bottom of the canopy

must also be chosen, since LAI is a function of vegetation, and not the ground return, but the ground return is used in accounting for energy after each voxel layer. The LAI is found by approximating an integral with a sum of LADen, scaled by the voxel height:

$$LAI = \int LADen \, dz \approx \sum_{i=V_b}^{V_t} LADen_i \, \Delta z \qquad (2)$$

where $V_b$ is the bottom vegetation voxel, $V_t$ is the top vegetation voxel, and $\Delta z$ is the voxel height. A $k$ value then is found by dividing the uncalibrated LAI value by the reference LAI. We investigated the LADen algorithm with a simulated scene in DIRSIG with a 10 m thick vegetation layer, raised 2 m above the ground with 10 cm leaf disks, uniformly distributed, a spherical leaf angle distribution, and an LAI = 4. We discovered that a ground return with same reflectance as the vegetation is needed for the LADen algorithm to work correctly, so all energy is accounted for and not "lost". In practice on real lidar waveforms, the ground return would need to be scaled to correct for the ratio of the ground-to-vegetation reflectance. If there is no ground return, or if it is very small, the error will be minimal, i.e., very little pulse power makes its way to the ground. We also found that the $\Delta z$ value has a significant effect on the estimated LADen. To illustrate this, various LADen profiles with different $\Delta z$ (m) are plotted in Figure 9 for a lidar system with a 5 m diameter footprint.

There are some noticeable differences between the $\Delta z$ curves. A sawtooth pattern for $\Delta z = 0.5$ m is the result of partial pulses being within voxels. Range resolution is half the pulse length, and the pulse length here is 3 ns, or ~0.9 m for a round trip, or ~0.45 m in range. Therefore, it is suggested that each voxel should be at least $2\Delta R$, with $\Delta R$ being the range resolution—this guarantees that an entire pulse is contained within a voxel. We expect the density to be constant within the canopy, but variations exist due to randomness in the canopy, and partially vegetation-filled voxels at the top and bottom of the canopy. In addition, "multibounce" returns will have an effect, as they cause a delay in the returned signal.

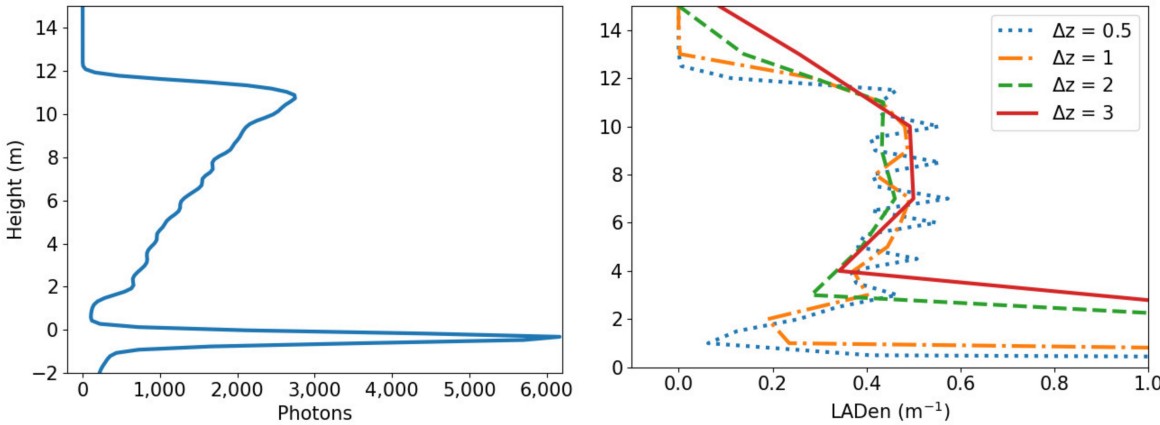

**Figure 9.** Lidar waveform, seen on the left, and the LADen profiles estimated using the waveform intensity data on the right.

Another approach to find the $k$ value is to fit a line to LAI data points, from varying reference LAIs, with the LAI estimated from lidar data using a value of one for the extinction. The slope of the line then is the extinction coefficient [54]. We tested this by making multiple vegetation layers with different LAI values, simulating the waveforms in DIRSIG, and investigating the increase in accuracy. In the investigation, we found a small bias, which can be accounted for to improve accuracy. The method in finding the extinction coefficient starts with finding uncorrelated LAI values for each vegetation layer, which are then plotted as a function of the reference LAI. Next, a best fit line is found, from which the slope is the extinction coefficient and the intercept is a bias. Ideally, no bias would

exist, as uncorrelated LAI = 0 when reference LAI = 0. The bias can be incorporated into LADen by dividing by *k* and the number of voxels used in the LAI calculation, and then subtracting the scaled bias from each LADen. The LAI is then recalculated with the LADen values that incorporate this bias. The plots, seen in Figure 10, display the accuracy in determining LAI. Note that the estimated LAI without subtracting the bias, lies below the one-to-one line. This is primarily due to the contribution of multibounce returns being delayed in the waveform, which makes the density voxels appear to be more transmissive than they truly are.

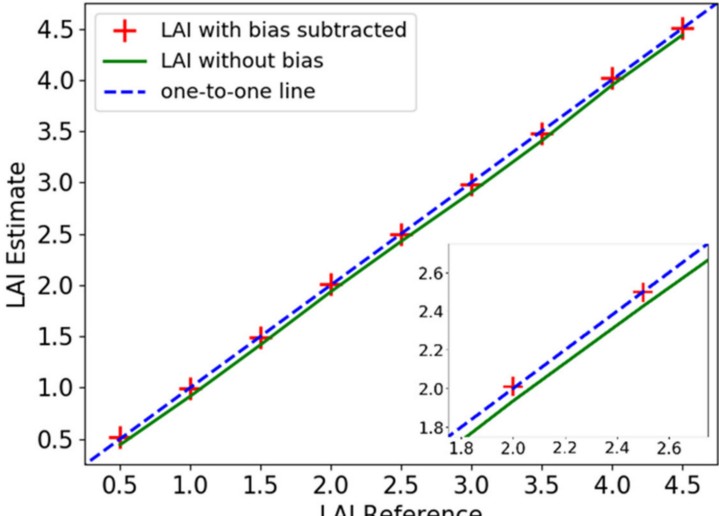

**Figure 10.** Accuracy of the slope method for finding the extinction coefficient, *k*. The plot compares the estimated LAI to the "truth" or reference LAI. Red crosses are the LAI estimates with the bias accounted for, while the solid green line shows where the estimates would lie without accounting for the bias (small zoomed insert shown, bottom-right). The blue dashed line is the one-to-one line.

When studying the sensitivity of the effect of leaf BSDF on lidar waveforms, we used the LADen from waveform method and then computed the associated LAI as a comparison metric, having only one reference waveform. Therefore, in our estimate of *k* we used a single reference value and assumed that the line goes through the origin to estimate *k*. Future studies could extend such analyses, comparable to that of Kamoske et al. [54] and Richardson et al. [94], to determine consistency across platforms and accuracy of the LADen from waveform method as compared to other methods.

## 2.7. Sensitivity Study Overview

BSDF effects were studied within DISIG by analyzing ALS waveform sensitivity to wavelength, footprint size, sensor view angle, and LAD. A 10 m thick abstract vegetation layer scene, raised two meters off the ground with an LAI = 4, was used as the primary scene. The scene consists of 10 cm diameter leaf disks with a uniform, random spatial distribution, from which different LADs were applied. Existing tree models in DIRSIG that closely resemble actual tree structure were then leveraged [95] to determine the impact of more natural scene geometry. The LAI and LAD were extracted to compare the results to what was found for the abstract vegetation layer.

The sensitivity study was accomplished by applying different permutations of wavelength, footprint size, view angle, and leaf angle distributions. Scattering properties were applied to the leaf disks with a sweetgum (*Liquidambar styraciflua*) leaf BSDF model (model) [86,87], Lambertian BSDF (Lambertian), leaf model reflectance with no transmittance (model-opaque), and leaf model reflectance with Lambertian transmittance (model-Lambertian) as seen in Figure 11.

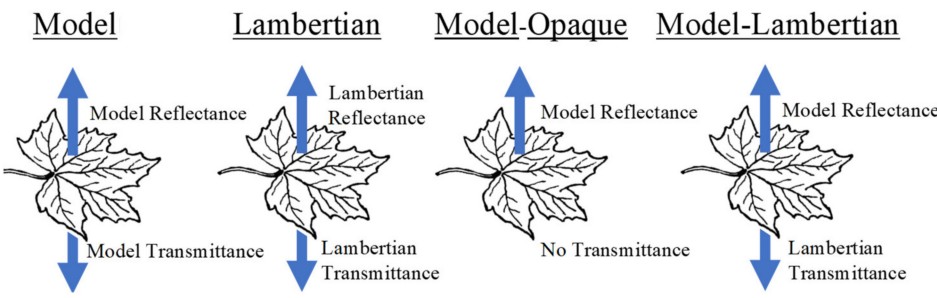

**Figure 11.** Visualization of the four different leaf BSDF configurations used for the sensitivity studies.

Common lidar system wavelengths of 550, 1064, and 1550 nm were included in the study. We chose the 550 nm wavelength (vs. the frequency doubled Nd:YAG wavelength of 532 nm often used in bathymetry) to correspond to the green peak reflection of vegetation. Several multispectral terrain classification lidar systems via a tunable laser have been developed at or close to this wavelength [96–98]. The chosen footprint diameters, 0.1, 0.5, and 5 m also encompass the range of current ALS systems. In order to vary the FOV, the focal length was changed to keep the FOV twice the diameter of the beam. Because the lidar beam shape is generally Gaussian, with a width defined at one sigma (60.6% of the peak height, 68.2% of the energy) [84], significant energy fills the entire FOV. The spherical, planophile, and plagiophile classical leaf angle distributions, commonly found for large trees [91,99], also were applied. Specific parameters investigated are listed in Table 3, from which all permutations were carried out (324 total simulations). The specific DIRSIG settings for the study are shown in Table 4. An important discovery was that the simple rad solver in DIRSIG4 cannot be used for specifying transmission, as it will collect multiple points along each ray, instead of applying transmission properties to the first "hit". Due to the nature of vegetation scenes that contain hundreds of thousands of facets, a manual search radius had to be implemented from which density and vector directions are calculated when interrogating the photon map. We set this search radius to 1 cm to ensure that it falls within the leaf geometry.

**Table 3.** List of parameters and associated values investigated for vegetation layer sensitivity study.

| Parameter | Values | | |
|---|---|---|---|
| Wavelength | 550 nm | 1064 nm | 1550 nm |
| Footprint Diameter | 0.1 m | 0.5 m | 5 m |
| Zenith View Angle | 0° | 22.5° | 45° |
| LAD | Spherical | Planophile | Plagiophile |
| BSDF | Model | Lambertian | Model-Opaque | Model-Lambertian |

**Table 4.** DIRSIG lidar settings for BSDF sensitivity study.

| Parameter | Value | Parameter | Value |
|---|---|---|---|
| Altitude | Varied (1000 m distance to target) | Gate Range | $6.55 \times 10^{-6}$ to $6.75 \times 10^{-5}$ s |
| Wavelength | 550, 1064, 1550 nm | Bin Size | 1 ns |
| Laser Spectral Width | $1 \times 10^{-5}$ μm | Receive Radius | 0.025 m |
| Pulse Energy | 0.2 mJ | Detector length | 250 μm square |
| Beam Shape | Gaussian | Focal Length | 1.25, 0.25, 0.025 m |
| Beam Divergence | $1 \times 10^{-4}, 5 \times 10^{-4}, 5 \times 10^{-3}$ m | Spatial subsampling | $101 \times 101$ |

**Table 4.** *Cont.*

| Parameter | Value | Parameter | Value |
|-----------|-------|-----------|-------|
| Pulse Length | 3 ns | Maximum events in photon map per pulse | 400,000 |
| Temporal Pulse Shape | Gaussian | Maximum source bundles per pulse | 800,000 |
| PM search radius | 1 cm | Maximum bounces per photon bundle | 10 (for multiscatter) |

## 2.8. Waveform Comparison Metrics

Three comparison metrics, namely percent increase, waveform overlap, and a novel method we call "LAI-from-waveform" were used in determining the effect that simplified leaf BSDF assumptions have on the lidar waveform. The waveform generated from the leaf model BSDF served as the reference for each metric, while the three remaining BSDF configurations (Lambertian, model-opaque, model-Lambertian) at the same wavelength, footprint, zenith angle, and LAD were the tested waveforms. By completing these comparisons, the metrics highlight the lidar configurations where the simplified leaf BSDF assumptions are or are not valid. The percent increase was calculated by $\frac{100*(Test-Reference)}{Reference}$ and the LAI from waveform was calculated as already described in Equation (2). The waveform overlap metric is defined as the intersection divided by the union of the 95% Poisson confidence interval about each waveform [95]. The metric assumes that DIRSIG generates the mean signal, variance can be described by a Poisson distribution (as waveforms are generated through MCRT), and other noise is negligible. Mathematically, this can be written as a function $O(w^i, w^j)$ in terms of waveforms $w^i, w^j$, a binary operator, $\Psi$, defining the confidence interval space in terms of the probability of seeing signal $S_b$, given a signal $w_b$ within a range bin, for $\alpha$ parameter (0.05 used here for a 95% confidence interval).

$$O(w^i, w^j) \equiv \frac{\sum_b \int_{S_b}^{\infty} \left[\Psi\left(S_b \middle| w_b^i, \alpha\right) \cap \Psi\left(S_b \middle| w_b^j, \alpha\right)\right] dS_b}{\sum_b \int_{S_b}^{\infty} \left[\Psi\left(S_b \middle| w_b^i, \alpha\right) \cup \Psi\left(S_b \middle| w_b^j, \alpha\right)\right] dS_b}. \tag{3}$$

The overlap metric encompasses the range (0,1), i.e., 0 for no agreement between the waveforms and 1 for total agreement. Due to a dependence on overlap area, more sensitivity is seen when compared to other metrics, mainly due to a steeper fall-off from the point of agreement [95].

## 2.9. Maple and Oak Grove Scene

Existing tree 3D models, previously created by Romanczyk [95], were used to create a small maple (*Acer rubrum*) and oak (*Quercus rubra*) grove to test BSDF effects on more realistic geometry that may be expected to exist naturally. The trees were randomly placed within the scene to have a maximum radii overlap of 40% with neighboring trees. The radii were determined as the distance from the center of the tree model to the outermost facet, also belonging to the tree in the x–y plane. Seven different tree models were randomly selected for creation of the grove, four of which were red maple and three were red oak trees. A description of the scene is shown in Figure 12, displaying the tree footprints, locations, model labels, and a thumbnail of each tree model.

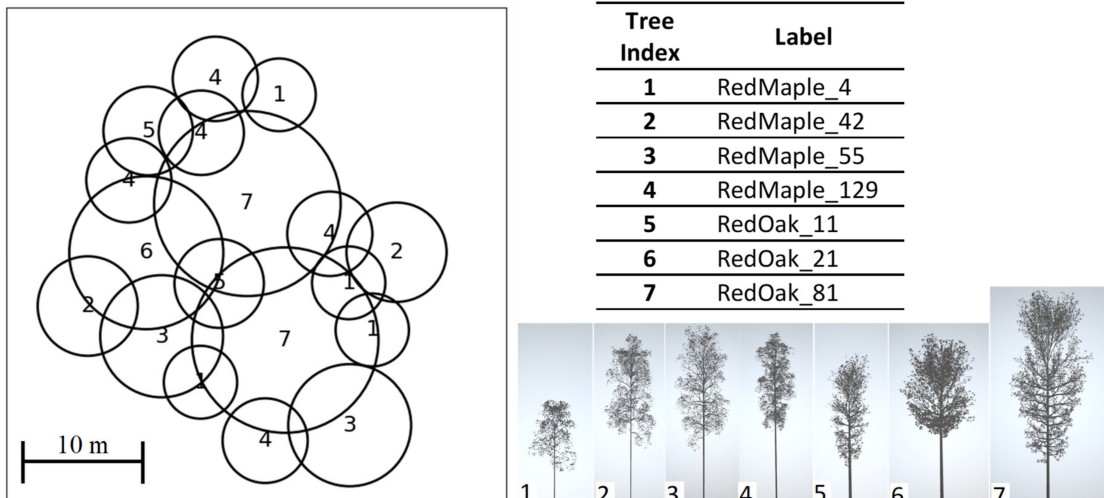

| Tree Index | Label |
|---|---|
| 1 | RedMaple_4 |
| 2 | RedMaple_42 |
| 3 | RedMaple_55 |
| 4 | RedMaple_129 |
| 5 | RedOak_11 |
| 6 | RedOak_21 |
| 7 | RedOak_81 |

**Figure 12.** Diagram of the construction of the maple and oak grove. On the left side of the figure is a map with locations and footprints of the tree models in the scene. The numbers correspond the tree model index, which is listed in the table on the right side of the figure. Under the table are thumbnail views of each tree model to provide a relative approximation of the shape and size of each tree.

Unlike the uniform vegetation layer, the maple and oak grove has different geometries and spatial statistics throughout the scene. We therefore interrogated at scene center and nine other random locations within a 10 m radius, centered over the canopy, in order to limit the geometry variability on our assessment of leaf BSDF effects. A DIRSIG-rendered red-green-blue (rgb) image of the maple and oak grove from directly overhead is seen in Figure 13, with the interrogation site locations indicated (a side view rgb image is also shown for perspective). Notice there is no ground level vegetation and material clutter that would be present in a natural environment. We limited our scene to trees, because we wanted to focus on lidar effects within the canopy, and not introduce other geometries that may impact computations and associated assessments.

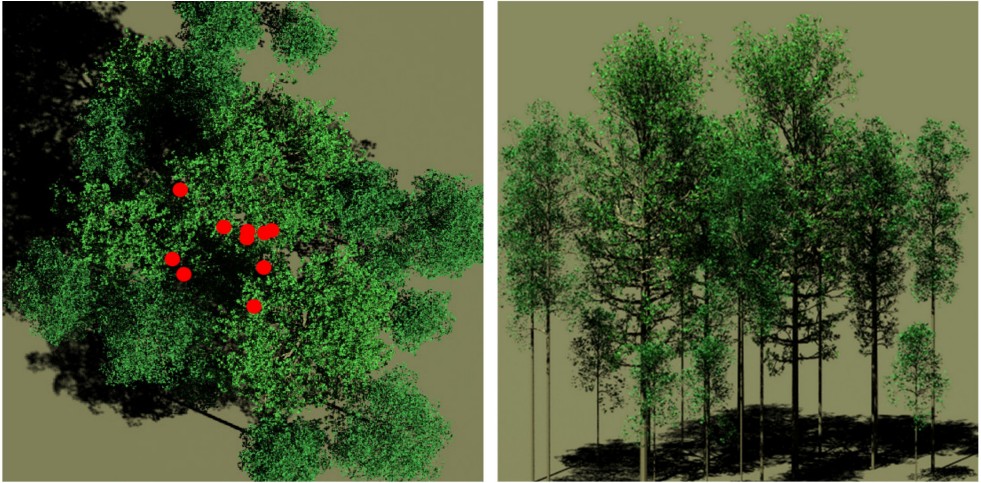

**Figure 13.** Images of maple and oak grove from directly overhead, (**left**), and from the side, (**right**). Red scatter points are displayed on the overhead image at locations where the canopy was interrogated.

In performing sensitivity studies on the maple and oak grove, we did not change leaf geometries, and instead analyzed the existing LAI and LAD within the scene in one meter pixels. The LAI was found by summing the one-sided area of leaf facets that fell within each one meter pixel. For each 1 m$^2$ cell, histograms of leaf angle were also created in 5° increments of leaf facet zenith angles.

We weighted each leaf angle contribution by the leaf facet area, since leaves consisted of more than one facet. This creates a histogram normalized by the LAI for each of the 1 m$^2$ cells. The histogram was then normalized to one and the mean leaf angle found by using the histogram bin values as weights for each corresponding bin leaf angle. Heat maps of the LAI and mean leaf angle over the scene are shown in Figure 14. The cumulative distribution function (cdf) for the LAD was also calculated via a cumulative sum over the normalized histogram values. We plotted the cdf for each of the pixels corresponding to the interrogation sites, shown in Figure 14. The planophile and spherical distributions are included as a comparison to the distributions that were used for the 10 m vegetation layer. Of the LADs implemented with the uniform vegetation layer, the spherical distribution is the closest to the LADs appearing in the grove scene, and we expect BSDF effects found with the grove scene to be somewhat similar to those results. The lidar interrogation coordinates, matching those on the heat maps in meters, corresponding pixel LAI, and mean leaf angle also are shown in Table 5.

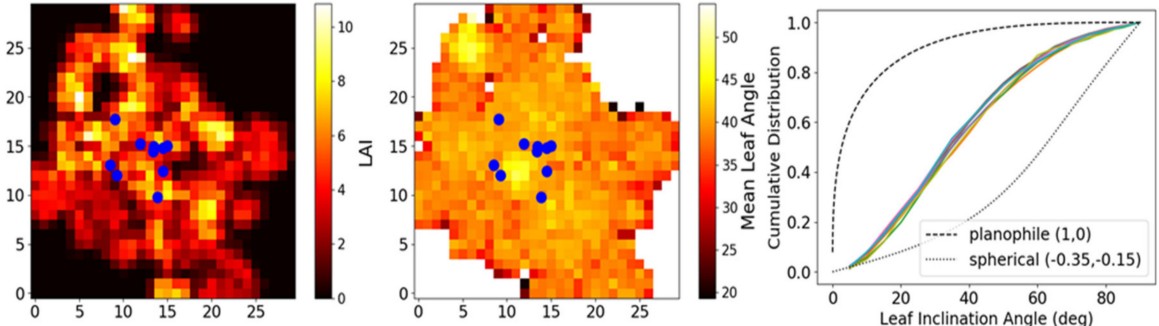

**Figure 14.** Heat map of LAI, left, and mean leaf angle, center, in one meter pixels. Blue scatter points are displayed on the heat maps for the lidar interrogation sites. Right, the cumulative distribution function curves for the 10 interrogation sites are plotted over each other, along with planophile and spherical distribution curves for reference.

**Table 5.** List of interrogation locations in x, y coordinates, LAI, and mean leaf angle.

| Loc # | X | Y | LAI | Mean Leaf Angle |
|-------|------|-------|------|-----------------|
| 0 | 15 | 15 | 5.61 | 38.9° |
| 1 | 14.5 | 15 | 4.14 | 41° |
| 2 | 8.52 | 12.43 | 2.58 | 40.8° |
| 3 | 13.45 | 13.03 | 4.09 | 39.2° |
| 4 | 9.04 | 14.92 | 5.31 | 40° |
| 5 | 14.54 | 17.75 | 5.61 | 39.9° |
| 6 | 13.86 | 14.82 | 6.08 | 38.9° |
| 7 | 11.91 | 9.81 | 2.82 | 39.2° |
| 8 | 9.29 | 15.22 | 1.02 | 40.6° |
| 9 | 13.39 | 14.46 | 4.09 | 39.2° |

DIRSIG lidar settings for the maple and oak grove simulations were kept to those shown in Table 4, except that simulations were only performed for 550 and 1064 nm, while the range gate was expanded to capture the larger vertical canopy. We limited the scans to a smaller subset of parameter permutations than those used on the uniform vegetation scene by performing all simulations at zero zenith and retaining the native leaf geometry of the trees. The simulation parameters from which all permutations were run are shown in Table 6.

**Table 6.** List of parameters and associated values for simulations of the maple and oak grove scene.

| Parameter | Values |
| --- | --- |
| Wavelength | 550 nm, 1064 nm |
| Footprint Diameter | 0.1 m, 0.5 m, 5 m |
| Zenith View Angle | 0° |
| BSDF | Model, Lambertian, Model-Opaque, Model-Lambertian |

## 3. Results

### 3.1. Vegetation Layer Results

The results of the vegetation layer sensitivity study are first presented as waveform plots for the zero zenith look angle and spherical leaf angle distribution at each of the three beam diameters and wavelengths seen in Figure 15. When evaluating the general waveform shapes, the smaller 0.1 m footprint exhibited more vegetation structure and detail, while the 0.5 m waveform smoothed out some of this detail, and the larger 5 m footprint yielded the expected result of almost a linear response through the canopy, due to the integration over many more objects in the FOV. Differences between the model and Lambertian leaf waveforms were observed, most pronounced at 550 nm. With the Lambertian leaves, the reflected energy was spread evenly into the hemisphere, while also accounting for the projected area. For the model leaves, much of the reflected energy was contained in the specular lobe at 550 nm and with a spherical leaf angle distribution, the majority of leaves are angled, which caused specular reflection away from the receiver. At wavelengths of 1064 and 1550 nm, leaves proved to be largely diffuse, showing less of a difference between the Lambertian and model leaves. Comparing the model-opaque to the model leaf at the various footprints and wavelengths revealed that at the 0.1 and 0.5 m footprints, the difference was fairly minimal. However, the difference is more significant at the 5 m footprint, most notable at the 1064 and 1550 nm wavelengths where multiscatter is a larger contribution. The greater differences observed at the large-footprint NIR wavelengths is a result of a larger FOV capturing more scattering events and increased transmission contributing to multiscatter. An examination of the difference between the model and the model-Lambertian leaf showed no distinguishable difference for any of the configurations.

We display the five highest and lowest values for each leaf BSDF simplification type and for each metric as bar charts in Figure 16 after applying the three comparison metrics. At first glance, we see that the Lambertian and model-opaque BSDF types both can create significant error. However, the model-Lambertian leaves exhibited very little effect, with a maximum percent increase of just over 1%, a minimum overlap of over 0.75, and the greatest LAI difference of −0.06. In comparison, the greatest overall errors seen for all permutations for each metric was a maximum percent increase of 80% observed for the Lambertian leaf, a minimum overlap of less than 0.05 for the model-opaque leaf, and an LAI increase of over 0.9 for the Lambertian leaf. The bar graphs also reveal trends in wavelength, footprint size, view zenith, and LAD for the various metrics and BSDF types. Interestingly, although some of the same configurations created the most error between the three metrics, many were different due to the different features that each of the metrics are sensitive to in the waveform. For example, the percent increase statistic only considers the sum difference of intensity magnitudes, while the overlap statistic examines how closely the waveforms align, and the LAI increase is sensitive to relative magnitudes between the upper and lower portions of the waveform. In fact, the LAI metric is insensitive to pure shifts in magnitude, specifically highlighting when ground returns are diminished. A large LAI increase, for instance, occurred for the model reflectance and no transmittance leaves at 1064 nm, with a 5 m footprint, 0° zenith view, and planophile leaf angle distribution. The ground returns were significantly affected by the opaque leaves, because the flat angled, non-transmitting leaves block much of the energy going to or coming from the ground, thus causing a significantly reduced ground return. Because of the small ground return, the LAI algorithm estimated a much

denser canopy, greatly overestimating the LAI. Some trends in the percent increase metric are that all the top Lambertian BSDF differences occurred at a 550 nm wavelength. As discussed earlier, this is due to the dominance of the specular lobe at this wavelength, which creates a significant difference in magnitude when reflected away from the lidar platform. All of the model reflection, non-transmitting leaves' largest percent increase differences occurred at 1064 nm, with a 5 m footprint, where multiscatter becomes a significant factor. Similar trends are also seen for the overlap statistic, where most similarities for the Lambertian leaf were at 1064 nm and the largest differences at 550 nm wavelengths. While the largest differences for the non-transmitting leaf were at 1064 nm, and the most similarities were observed at 550 nm.

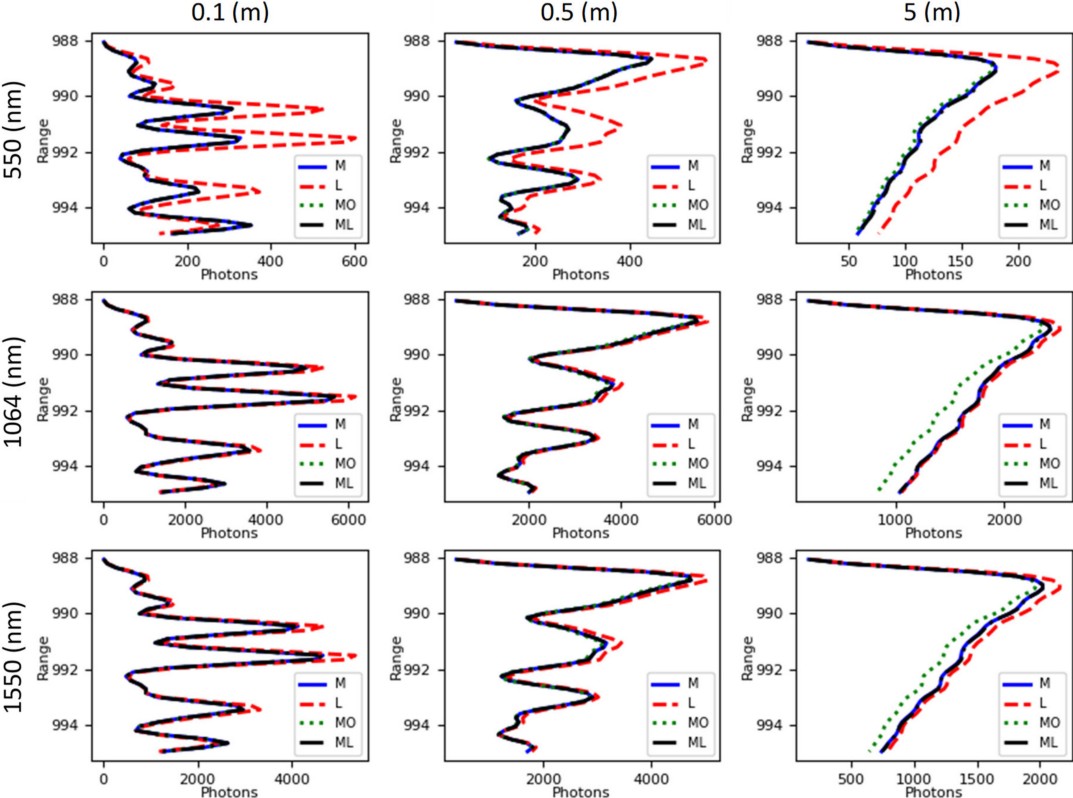

**Figure 15.** Waveform comparison plots for each leaf BSDF type at the zero zenith angle and spherical leaf angle distribution of leaves. Plots are shown for the three beam footprint diameters of 0.1, 0.5, and 5 m (left-to-right), and the three wavelengths of 550, 1064, and 1550 nm (top-to-bottom). The legend symbols for the four leaf BSDF types are "M" for model, "L" for Lambertian, "MO" for model reflectance and no transmittance, and "ML" for model reflectance and Lambertian transmittance. In order to display more detail, the last five meters of range are excluded, thus only displaying the canopy returns.

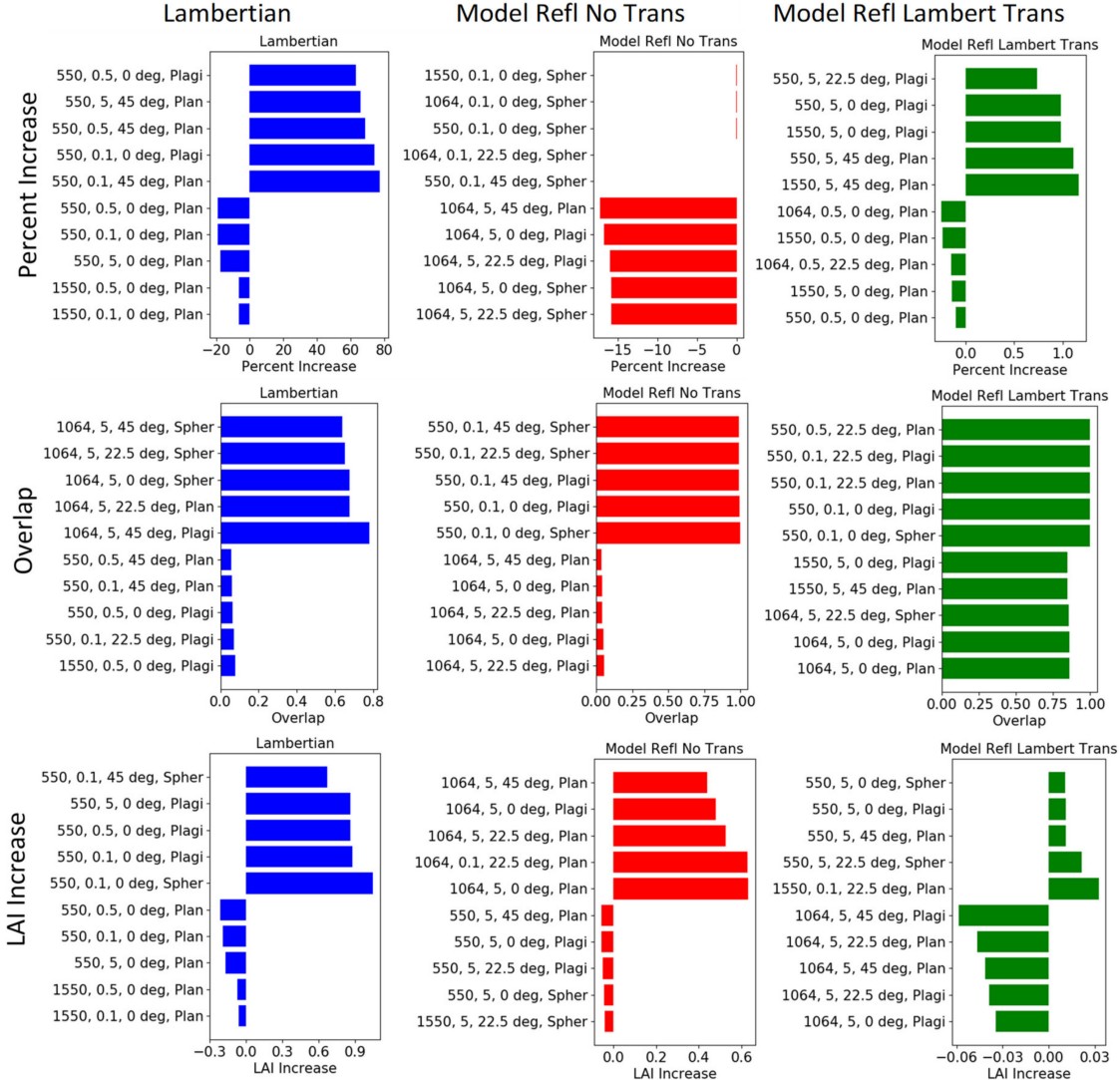

**Figure 16.** Bar graphs of the five highest and lowest values (top five and bottom five bars in each plot, respectively) for each leaf BSDF simplification type, left-to-right: Lambertian, model-opaque, and model-Lambertian; each metric, top-to-bottom: percent increase, overlap, and LAI increase. Note, the axes are scaled differently for each graph as the scales between the different leaf types are drastically different for many of the metrics. Labels for the simulations are wavelength (nm), footprint size (m), zenith angle (degree), and LAD type (spherical (Spher), planophile (Plan), and plagiophile (Plagi)).

Each metric for every permutation is shown in the Appendix A for completeness (Tables A1–A3). The top 10% values are highlighted in yellow, while the bottom 10% are shown in blue to make the greater offenders more easily identifiable (except for the overlap statistic). These tables are useful for simulations of specific lidar systems and understanding possible error due to leaf BSDF simplifying assumptions. One reason to use simplified leaf BSDF properties is that computational time can be greatly increased. For instance, if transmission through the leaves is not a significant contributor for a specific configuration, eliminating transmission rays can result in significant computational efficiencies. It also is valuable to know if Lambertian assumptions are valid for a specific configuration, since limited data exist on individual leaf BSDF estimates from actual measurements. For example, the NEON AOP [30] operates at 1064 nm, typically with a 0.5 m footprint and a scanning angle of 0°–30°. We may want to know what error results when simulating Lambertian leaves in a radiative transfer model for the NEON AOP, assuming spherical LAD and a 22.5° view zenith angle. We can conclude from each table that the percent difference is expected to be under 4%, the overlap 0.36, and the LAI difference less

than 0.1. These errors most likely are tolerable, and modeling Lambertian leaves is a solid assumption for the NEON AOP platform.

### 3.2. Maple and Oak Grove Scene Results

We investigated more realistic geometry, as is present naturally in a forest canopy, for the maple and oak grove sensitivity study. Major differences, when compared to the abstract uniform vegetation layer previously used, included different materials (e.g., leaves, trunks, stems, and branches), leaf geometries, and leaf distributions. As previously mentioned, because of the varying statistics in the scene, ten locations were interrogated for all permutations of parameters listed in Table 6. The resulting waveforms from location zero (center scene) are shown in Figure 17, and exhibited similar trends as those seen with the vegetation layer spherical LAD from Figure 15. We noted no distinguishable difference between the model leaf and the model reflectance Lambertian transmittance leaf (model-Lambertian) for all cases, except for a very slight increase at the 1064 nm wavelength and 5 m footprint. The largest differences were observed with the Lambertian leaf, most prominent in the 550 nm wavelength, 5 m footprint plot. There was also a significant decline in intensity from the non-transmitting leaf in the 1064 nm, 5 m plot.

A further evaluation was completed by using the model leaf as a reference and making comparisons with each of the three metrics, namely percent increase, waveform overlap, and LAI from waveform, in the same manner as was previously accomplished with the abstract vegetation layer simulations. However, to capture the variability due to the varying materials and geometries in a realistic canopy, we evaluated the statistics resulting from the ten different interrogation locations over the canopy. This was accomplished by producing box plots for each of the three metrics, shown in Figure 18. Each box encompasses data within the 25th–75th percentile, also known as the interquartile range (IQR), the whiskers are drawn at 1.5 times the IQR to the outermost data point, and outliers beyond 1.5 times the IQR are separately plotted as small circles. Figure 18 shows that there were several outliers, some fairly extreme. This was attributed to the photon mapping and search radius method having difficulty capturing the geometry accurately. First, in terms of the percent increase metric, as expected, the largest differences can be observed for the Lambertian BSDF at 550 nm, which was attributed to the specular contribution. The variability increased for the smaller footprints, also resulting in data points with a larger difference, the maximum representing an almost 30% increase. The model-opaque BSDF exhibited little difference in percent increase for the 550 nm wavelength, but showed as much as a 15% decrease for the 1064 nm with a 5 m lidar footprint. Lastly, the model-Lambertian waveforms showed almost no difference in intensity, although a few simulations resulted in outliers. The next statistic we looked at with box plots was the waveform overlap metric, shown as the second row in Figure 18. Overall, there was a larger spread of data points due to the higher sensitivity and faster fall-off of the metric. Again, large differences (data points closest to zero) were seen for the 550 nm wavelength, Lambertian BSDF. Significant differences also were seen for the 1064 nm wavelength and model-opaque leaves, the 5 m footprint data being the largest offenders. The 1064 nm Lambertian data, specifically the 0.1 m footprint, also exhibited a substantial dissimilarity. The lower overlap, evident for the 1064 nm wavelength Lambertian BSDF that is not registered by the percent increase statistic, suggests that the waveform can be considerably different, even though the summed intensity as compared to the model BSDF may be similar. The last metric we evaluated, namely LAI from waveform, had to be converted to a relative value, because the LAI varied between each lidar interrogation location. We therefore report the LAI from waveform as a percent increase. On first observation, the 550 nm wavelength Lambertian data exhibited the greatest error with the largest data point at just under 15%. The only other configuration with significant change was the model opaque BSDF at 1064 nm for the 5 m lidar footprint, resulting in an approximately 6% decrease in estimated LAI. Note that some of the LAI metric means are below the box and whiskers due to an extreme outlier. The extreme outliers are due to the different geometries in the scene, causing a single interrogation site to result in poor LAI correlation.

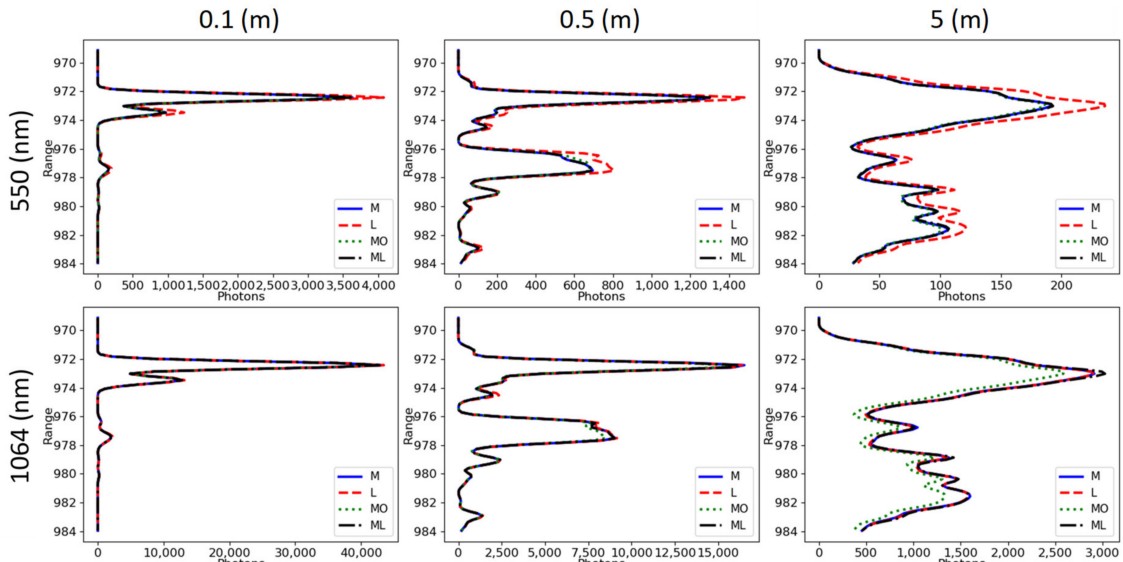

**Figure 17.** Waveform comparison plots for the maple and oak grove scene for each leaf BSDF type. Plots are shown for the three beam footprint diameters of 0.1, 0.5, and 5 m (left-to-right) and the two wavelengths of 550 and 1064 nm (top-to-bottom). The legend symbols for the four leaf BSDF types are "M" for model, "L" for Lambertian, "MO" for model reflectance and no transmittance, and "ML" for model reflectance and Lambertian transmittance. Only the main canopy range is shown, thereby highlighting any differences.

In summary, evaluations with a scene consisting of natural geometries revealed the same trends as were shown with the abstract vegetation layer, suggesting that abstract scenes are effective in exploring the impacts that scene properties have on lidar returns. The largest differences between the abstract scene and the maple and oak grove scene were a result of the increased variability in waveform shapes and magnitudes from the inconsistency in scene geometry. Variability in scene geometry resulted in the spread of metric data from the 10 different interrogation sites depending on the configuration and applied metric, which implied the necessity of evaluations at multiple scene locations. Simulations and ensuing comparisons revealed that simplifying BSDF assumptions are valid at certain configurations, depending on the error tolerance and specific application. As a result, the impact that leaf BSDF had on lidar returns, both in terms of magnitudes and shapes was quantified, and an awareness of possible effects to subsequent data processing is gained.

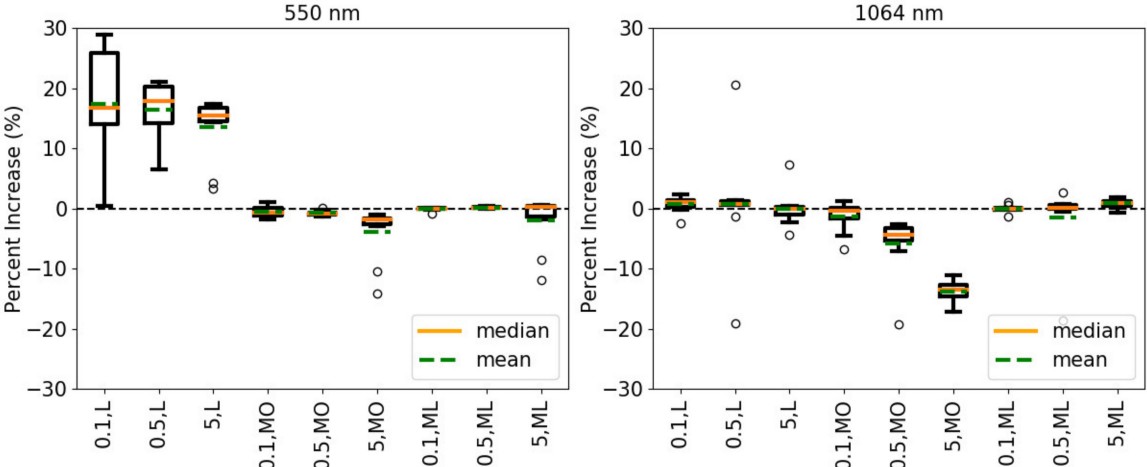

**Figure 18.** *Cont.*

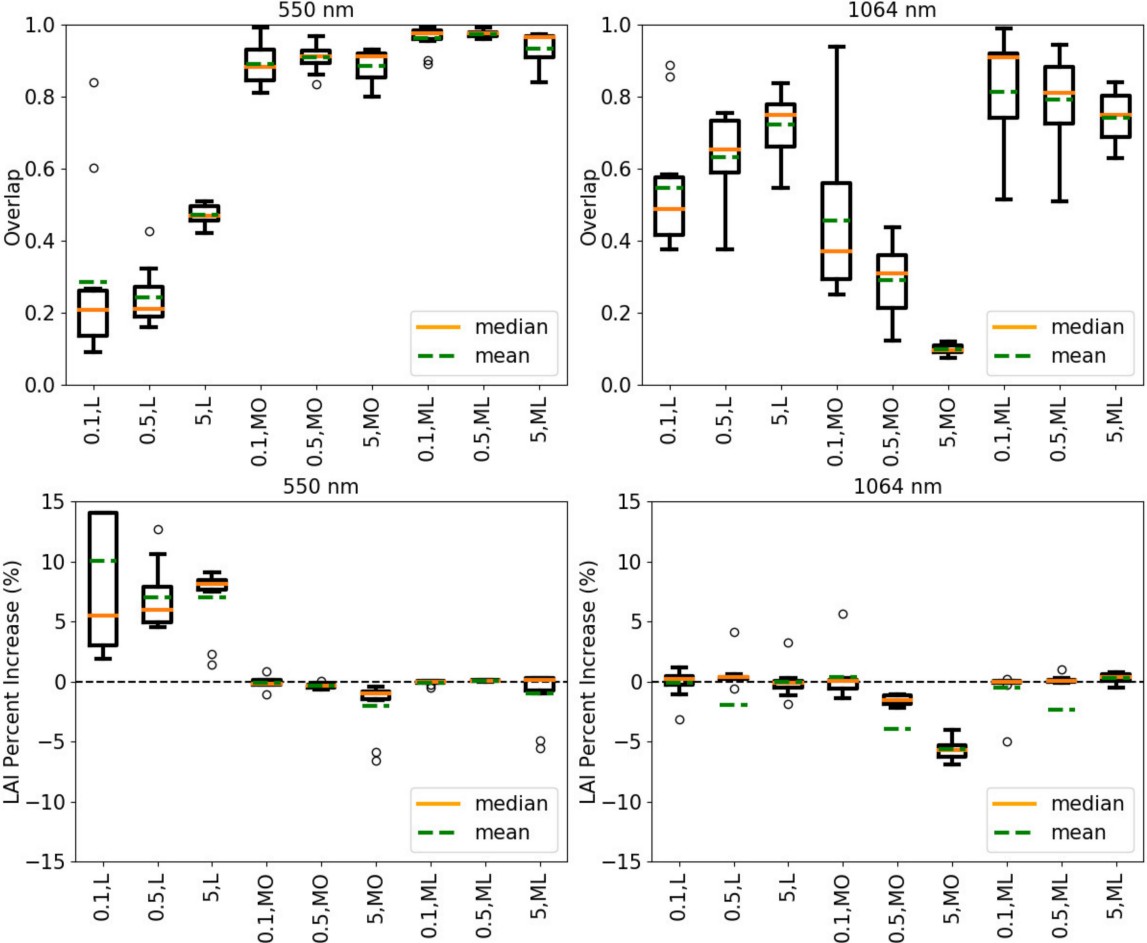

**Figure 18.** Box plots of the percent increase, waveform overlap, and LAI percent increase metric (rows top-to-bottom) for the 550 nm wavelength (left column) and the 1064 nm wavelength (right column). The plots encompass the ten interrogation sites for each of the three lidar footprints (0.1, 0.5, and 5 m) and three optical properties (Lambertian, L; model-opaque, MO; and model-Lambertian, ML).

## 4. Conclusions

The DIRSIG radiative transfer model was used to study the effect that individual leaf optical scattering properties have on lidar waveforms, as compared to traditional Lambertian and non-transmitting scattering assumptions. Validation comparisons were first completed in order to gain confidence in simulated lidar waveforms produced by DIRSIG, specifically for the RAMI HET17 [83] scenario and multiscatter contribution, previously shown with the DART model [39]. DIRSIG simulations compared well against these scenarios, displaying the same general trends, effects, and magnitudes. Sensitivity studies were then accomplished by applying a leaf BSDF model [86,87] to leaf facets, with three leaf BSDF assumptions: Lambertian scattering, non-transmitting, and model reflectance with Lambertian transmission. Three metrics were used to highlight differences, namely percent increase, waveform overlap, and a novel method we call "LAI-from-waveform". An abstract vegetation layer, as well as a realistic maple and oak grove based on 3D tree models, were used as the modeled scenes. Each metric was able to identify the effects of different model assumptions on the resultant waveform. The percent increase statistic showed magnitude differences, while the LAI metric revealed asymmetries between the top and bottom of the waveform. The waveform overlap metric largely followed errors highlighted in the LAI metric, while also revealing differences in the actual shape, noticeable for the NIR wavelength, small-footprint FOV, and non-transmitting leaves. In general, the model reflectance Lambertian transmittance assumption was seen to represent

a solid approximation with minimal error. Performing simulations with non-transmitting leaves produced reasonably accurate waveforms for all 550 nm wavelength scenarios, and NIR wavelengths with small footprints. Conversely, significant error was observed with non-transmitting leaves for NIR wavelength and large-footprint configurations. Simulations with leaves that are both Lambertian reflecting and transmitting showed that this assumption is valid for lidar systems operating in the NIR, where the leaf BSDF specular lobe is not a large contributing factor. However, significant errors can be incurred if Lambertian leaf assumptions are made for lidar systems operating at visible wavelengths. We ultimately were able to obtain a better understanding of valid scattering assumptions, light interactions within the forest canopy, and methods to produce higher fidelity simulations by quantifying the impacts that individual leaf BSDF had on simulated waveforms. Such improved radiative transfer modeling paves the way for the development of next generation remote sensing systems and data processing algorithms, enabling accurate and precise forest structural assessments.

**Author Contributions:** Conceptualization, methodology, validation, analysis, and writing—original draft preparation, B.D.R.; software, A.A.G., S.D.B., and M.G.S.; writing—review and editing, J.A.v.A.; conceptualization, K.K. All authors have read and agreed to the published version of the manuscript.

**Funding:** This research received no external funding.

**Acknowledgments:** The authors would like to thank the entire Digital Imaging and Remote Sensing Image Generation (DIRSIG) team past and present, notably Rolando Raqueno and Byron Eng for their contributions in making this work possible. We would also like to thank Jim Bodie and Brett Matzke for computer server support. We also acknowledge administrative support provided by Colleen McMahon and Melanie Warren.

**Conflicts of Interest:** The authors declare no conflict of interest.

## Appendix A

**Table A1.** Percent increase statistic for each configuration. The shorthand for the descriptions of the columns are wavelength in nm and footprint in m, while the rows are zenith view angle in degrees, LAD type (Sp–spherical, Plan–planophile, Plag–plagiophile), and leaf BSDF (L–Lambertian, MO–model reflectance non-transmitting, ML–model reflectance Lambertian transmittance). Yellow highlights are the top 10% while the blue highlights are the bottom 10% of the data points.

| Percent Increase | 550, 0.1 | 550, 0.5 | 550, 5 | 1064, 0.1 | 1064, 0.5 | 1064, 5 | 1550, 0.1 | 1550, 0.5 | 1550, 5 |
|---|---|---|---|---|---|---|---|---|---|
| 0 deg, Sp, L | 25.66% | 22.38% | 22.32% | 3.72% | 3.01% | 1.85% | 6.02% | 5.14% | 4.28% |
| 0 deg, Sp, MO | −0.04% | −0.58% | −2.61% | −0.07% | −1.67% | −15.85% | −0.09% | −1.50% | −8.74% |
| 0 deg, Sp, ML | 0.01% | 0.11% | 0.68% | 0.00% | −0.04% | 0.42% | 0.01% | 0.05% | 0.50% |
| 0 deg, Plan, L | −19.03% | −19.07% | −17.44% | −4.42% | −4.52% | −3.97% | −6.47% | −6.55% | −5.82% |
| 0 deg, Plan, MO | −0.10% | −0.90% | −2.24% | −0.32% | −2.94% | −13.12% | −0.29% | −2.47% | −7.98% |
| 0 deg, Plan, ML | −0.03% | −0.10% | −0.02% | −0.05% | −0.25% | −0.09% | −0.06% | −0.24% | −0.14% |
| 0 deg, Plag, L | 73.95% | 63.22% | 60.02% | 8.19% | 7.12% | 4.99% | 13.49% | 12.02% | 10.86% |
| 0 deg, Plag, MO | −0.10% | −0.96% | −3.63% | −0.30% | −2.41% | −16.70% | −0.26% | −2.08% | −9.61% |
| 0 deg, Plag, ML | 0.01% | 0.19% | 0.98% | −0.01% | 0.00% | 0.70% | −0.01% | 0.11% | 0.98% |
| 22.5 deg, Sp, L | 33.82% | 30.23% | 29.43% | 4.61% | 3.91% | 2.43% | 7.51% | 6.69% | 5.68% |
| 22.5 deg, Sp, MO | −0.13% | −0.75% | −3.00% | −0.01% | −1.90% | −15.83% | −0.20% | −1.71% | −8.78% |
| 22.5 deg, Sp, ML | 0.03% | 0.13% | 0.66% | −0.01% | −0.03% | 0.69% | 0.01% | 0.10% | 0.66% |
| 22.5 deg, Plan, L | 9.87% | 17.48% | 18.20% | 1.36% | 2.27% | 1.86% | 2.38% | 4.07% | 4.05% |
| 22.5 deg, Plan, MO | −0.23% | −1.24% | −3.05% | −0.71% | −3.38% | −14.18% | −0.57% | −2.83% | −8.71% |
| 22.5 deg, Plan, ML | 0.00% | 0.00% | 0.32% | −0.04% | −0.15% | −0.07% | −0.01% | −0.04% | 0.28% |
| 22.5 deg, Plag, L | 41.30% | 40.11% | 39.35% | 5.32% | 4.97% | 3.29% | 8.75% | 8.47% | 7.68% |
| 22.5 deg, Plag, MO | −0.18% | −0.93% | −3.38% | −0.10% | −2.40% | −15.97% | −0.28% | −2.13% | −9.23% |
| 22.5 deg, Plag, ML | 0.00% | 0.20% | 0.73% | −0.03% | −0.03% | 0.33% | 0.00% | 0.09% | 0.73% |
| 45 deg, Sp, L | 35.91% | 39.51% | 35.47% | 4.64% | 4.93% | 3.01% | 7.69% | 8.33% | 6.60% |
| 45 deg, Sp, MO | 0.00% | −0.70% | −2.89% | −0.15% | −1.79% | −15.29% | −0.23% | −1.60% | −8.81% |
| 45 deg, Sp, ML | −0.02% | 0.15% | 0.58% | 0.00% | −0.02% | 0.63% | −0.01% | 0.08% | 0.66% |
| 45 deg, Plan, L | 77.44% | 68.79% | 65.75% | 8.35% | 7.57% | 5.60% | 13.86% | 12.78% | 11.70% |
| 45 deg, Plan, MO | −0.59% | −1.41% | −5.11% | −0.94% | −3.05% | −17.24% | −0.91% | −2.64% | −10.83% |
| 45 deg, Plan, ML | 0.09% | 0.29% | 1.11% | −0.01% | −0.01% | 0.56% | 0.07% | 0.19% | 1.17% |
| 45 deg, Plag, L | 30.35% | 23.37% | 17.72% | 4.06% | 3.14% | 1.50% | 6.73% | 5.37% | 3.53% |
| 45 deg, Plag, MO | −0.11% | −0.71% | −2.70% | −0.12% | −1.93% | −14.32% | −0.18% | −1.70% | −8.28% |
| 45 deg, Plag, ML | 0.03% | 0.10% | 0.36% | −0.01% | 0.02% | 0.46% | 0.02% | 0.07% | 0.49% |

**Table A2.** Overlap statistic for each configuration. The shorthand for the descriptions for the columns are wavelength in nm and footprint in m, while the rows are zenith view angle in degrees, LAD type (Sp–spherical, Plan–planophile, Plag–plagiophile), and leaf BSDF (L–Lambertian, MO–model reflectance non-transmitting, ML–model reflectance Lambertian transmittance). Yellow highlights are the top 10% of the data points.

| Overlap | 550, 0.1 | 550, 0.5 | 550, 5 | 1064, 0.1 | 1064, 0.5 | 1064, 5 | 1550, 0.1 | 1550, 0.5 | 1550, 5 |
|---|---|---|---|---|---|---|---|---|---|
| 0 deg, Sp, L | 0.2301 | 0.1865 | 0.278 | 0.3254 | 0.4424 | 0.6758 | 0.2509 | 0.2572 | 0.4672 |
| 0 deg, Sp, MO | 0.9975 | 0.9555 | 0.8668 | 0.9761 | 0.639 | 0.0689 | 0.9769 | 0.6928 | 0.1593 |
| 0 deg, Sp, ML | 0.999 | 0.9934 | 0.9611 | 0.9945 | 0.98 | 0.8796 | 0.9959 | 0.9869 | 0.919 |
| 0 deg, Plan, L | 0.0881 | 0.1194 | 0.1773 | 0.147 | 0.1878 | 0.3409 | 0.1017 | 0.1314 | 0.2101 |
| 0 deg, Plan, MO | 0.9863 | 0.8966 | 0.8277 | 0.8712 | 0.3311 | 0.0416 | 0.8908 | 0.413 | 0.1224 |
| 0 deg, Plan, ML | 0.9965 | 0.9872 | 0.9882 | 0.9772 | 0.9137 | 0.8608 | 0.977 | 0.922 | 0.9271 |
| 0 deg, Plag, L | 0.0804 | 0.063 | 0.0842 | 0.1143 | 0.1397 | 0.3807 | 0.0913 | 0.0759 | 0.1397 |
| 0 deg, Plag, MO | 0.9918 | 0.9381 | 0.837 | 0.9099 | 0.5044 | 0.0503 | 0.9298 | 0.5902 | 0.1355 |
| 0 deg, Plag, ML | 0.999 | 0.989 | 0.9487 | 0.9905 | 0.9863 | 0.859 | 0.9947 | 0.9716 | 0.8451 |
| 22.5 deg, Sp, L | 0.2287 | 0.1662 | 0.2524 | 0.3179 | 0.3596 | 0.6534 | 0.2557 | 0.2153 | 0.4231 |
| 22.5 deg, Sp, MO | 0.9896 | 0.9483 | 0.8753 | 0.9462 | 0.6135 | 0.065 | 0.9487 | 0.6688 | 0.2145 |
| 22.5 deg, Sp, ML | 0.9982 | 0.9914 | 0.9679 | 0.9926 | 0.9841 | 0.8548 | 0.9957 | 0.9712 | 0.9019 |
| 22.5 deg, Plan, L | 0.0763 | 0.196 | 0.3274 | 0.2255 | 0.4595 | 0.6776 | 0.1092 | 0.2628 | 0.4516 |
| 22.5 deg, Plan, MO | 0.9732 | 0.891 | 0.8358 | 0.7504 | 0.3313 | 0.0425 | 0.8086 | 0.4088 | 0.1525 |
| 22.5 deg, Plan, ML | 0.9989 | 0.9986 | 0.9821 | 0.9824 | 0.9493 | 0.8881 | 0.9949 | 0.984 | 0.9285 |
| 22.5 deg, Plag, L | 0.0687 | 0.1121 | 0.1689 | 0.1472 | 0.2646 | 0.5686 | 0.0866 | 0.1448 | 0.2694 |
| 22.5 deg, Plag, MO | 0.985 | 0.9374 | 0.8562 | 0.9414 | 0.5179 | 0.0526 | 0.9205 | 0.5924 | 0.1833 |
| 22.5 deg, Plag, ML | 0.9987 | 0.9868 | 0.9675 | 0.9868 | 0.9805 | 0.893 | 0.993 | 0.9688 | 0.8923 |
| 45 deg, Sp, L | 0.1812 | 0.1275 | 0.2485 | 0.291 | 0.3128 | 0.6369 | 0.2158 | 0.1794 | 0.4123 |
| 45 deg, Sp, MO | 0.9871 | 0.9599 | 0.8986 | 0.856 | 0.6701 | 0.0766 | 0.9033 | 0.7213 | 0.2662 |
| 45 deg, Sp, ML | 0.99 | 0.9909 | 0.9777 | 0.9795 | 0.9613 | 0.8877 | 0.975 | 0.975 | 0.9159 |
| 45 deg, Plan, L | 0.061 | 0.0576 | 0.1055 | 0.1267 | 0.1331 | 0.3971 | 0.0822 | 0.078 | 0.1474 |
| 45 deg, Plan, MO | 0.9584 | 0.9191 | 0.8274 | 0.7537 | 0.4523 | 0.0361 | 0.7833 | 0.5442 | 0.1678 |
| 45 deg, Plan, ML | 0.9934 | 0.9825 | 0.9588 | 0.993 | 0.9814 | 0.8743 | 0.9808 | 0.9582 | 0.8481 |
| 45 deg, Plag, L | 0.1395 | 0.223 | 0.4765 | 0.307 | 0.4612 | 0.7817 | 0.1974 | 0.3022 | 0.6126 |
| 45 deg, Plag, MO | 0.9903 | 0.9505 | 0.8922 | 0.9053 | 0.625 | 0.0673 | 0.9438 | 0.6854 | 0.2666 |
| 45 deg, Plag, ML | 0.9985 | 0.9937 | 0.986 | 0.9941 | 0.9792 | 0.8785 | 0.994 | 0.9809 | 0.9223 |

**Table A3.** LAI statistic for each configuration. The shorthand for the descriptions for the columns are wavelength in nm and footprint in m, while the rows are zenith view angle in degrees, LAD type (Sp–spherical, Plan–planophile, Plag–plagiophile), and leaf BSDF (L–Lambertian, MO–model reflectance non-transmitting, ML–model reflectance Lambertian transmittance). Yellow highlights are the top 10% while the blue highlights are the bottom 10% of the data points.

| LAI | 550, 0.1 | 550, 0.5 | 550, 5 | 1064, 0.1 | 1064, 0.5 | 1064, 5 | 1550, 0.1 | 1550, 0.5 | 1550, 5 |
|---|---|---|---|---|---|---|---|---|---|
| 0 deg, Sp, L | 5.0463 | 4.5001 | 4.5566 | 4.1367 | 4.0651 | 4.0431 | 4.2234 | 4.1114 | 4.1026 |
| 0 deg, Sp, MO | 3.9983 | 3.9859 | 3.9569 | 3.998 | 3.9756 | 4.3485 | 3.9967 | 3.9685 | 3.9645 |
| 0 deg, Sp, ML | 4.0006 | 4.0029 | 4.0106 | 4 | 3.9987 | 3.9754 | 4.0003 | 4.0005 | 3.9872 |
| 0 deg, Plan, L | 3.808 | 3.7838 | 3.8298 | 3.9577 | 3.9517 | 3.9523 | 3.9373 | 3.9288 | 3.9431 |
| 0 deg, Plan, MO | 3.9991 | 3.9908 | 3.9854 | 3.9978 | 3.9794 | 4.6313 | 3.9975 | 3.9748 | 4.0366 |
| 0 deg, Plan, ML | 3.9998 | 3.999 | 4.0008 | 3.9996 | 3.9971 | 3.9723 | 3.9995 | 3.9972 | 3.9881 |
| 0 deg, Plag, L | 4.8775 | 4.8639 | 4.859 | 4.1038 | 4.0983 | 4.068 | 4.1699 | 4.1672 | 4.1393 |
| 0 deg, Plag, MO | 3.9983 | 3.9821 | 3.9439 | 3.9967 | 3.9735 | 4.4781 | 3.9966 | 3.9683 | 3.9795 |
| 0 deg, Plag, ML | 4.0001 | 4.0035 | 4.0109 | 4 | 3.9983 | 3.9656 | 3.9999 | 4.0013 | 3.9888 |
| 22.5 deg, Sp, L | 4.498 | 4.5845 | 4.6436 | 4.069 | 4.0739 | 4.0435 | 4.1126 | 4.1274 | 4.1095 |
| 22.5 deg, Sp, MO | 3.9978 | 3.984 | 3.9606 | 4.0007 | 3.9755 | 4.3375 | 3.9969 | 3.9684 | 3.9599 |
| 22.5 deg, Sp, ML | 4.0005 | 4.0036 | 4.0214 | 3.9995 | 3.9989 | 3.9722 | 4 | 4.0016 | 3.9839 |
| 22.5 deg, Plan, L | 3.9771 | 4.1693 | 4.1501 | 3.9522 | 4.0227 | 3.9962 | 4.0191 | 4.0401 | 4.0256 |
| 22.5 deg, Plan, MO | 3.9997 | 3.9863 | 3.972 | 4.6291 | 3.9745 | 4.5246 | 4.1428 | 3.9707 | 4 |
| 22.5 deg, Plan, ML | 3.9999 | 3.9999 | 4.0037 | 4.0084 | 3.9979 | 3.9532 | 4.0324 | 3.9994 | 3.9812 |
| 22.5 deg, Plag, L | 4.3665 | 4.5566 | 4.5504 | 4.0506 | 4.0702 | 4.0225 | 4.0828 | 4.1196 | 4.1018 |
| 22.5 deg, Plag, MO | 3.9981 | 3.9842 | 3.9503 | 4.0002 | 3.9755 | 4.3937 | 3.9973 | 3.9692 | 3.9754 |
| 22.5 deg, Plag, ML | 4 | 4.0034 | 4.0071 | 3.9997 | 3.999 | 3.9612 | 4 | 4.0007 | 3.9881 |
| 45 deg, Sp, L | 4.6692 | 4.5464 | 4.3977 | 4.0864 | 4.0697 | 4.0145 | 4.1431 | 4.118 | 4.0733 |
| 45 deg, Sp, MO | 4.0001 | 3.9884 | 3.9646 | 3.9977 | 3.9829 | 4.43 | 3.9957 | 3.9772 | 3.9842 |
| 45 deg, Sp, ML | 3.9995 | 4.0025 | 4.0034 | 4.0001 | 3.9991 | 3.9778 | 3.9998 | 4.0009 | 3.9912 |
| 45 deg, Plan, L | 4.6175 | 4.5557 | 4.432 | 4.0761 | 4.0673 | 4.0214 | 4.1247 | 4.1135 | 4.0776 |
| 45 deg, Plan, MO | 3.9936 | 3.9843 | 3.9431 | 3.9915 | 3.9795 | 4.4404 | 3.9912 | 3.9742 | 3.9687 |
| 45 deg, Plan, ML | 4.001 | 4.0033 | 4.0112 | 3.9998 | 3.9987 | 3.9588 | 4.0007 | 4.0016 | 4.0005 |
| 45 deg, Plag, L | 4.3773 | 4.2722 | 4.1884 | 4.0525 | 4.0375 | 3.9813 | 4.0865 | 4.064 | 4.035 |
| 45 deg, Plag, MO | 3.9984 | 3.9908 | 3.9719 | 3.9989 | 3.9858 | 4.3718 | 3.9976 | 3.9799 | 3.9948 |
| 45 deg, Plag, ML | 4.0004 | 4.0012 | 4.0037 | 3.9999 | 3.9992 | 3.9416 | 4.0002 | 4.0004 | 3.9913 |

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
