# Peer review of "Simulations of Leaf BSDF Effects on Lidar Waveforms"

_remotesensing, doi:10.3390/rs12182909_

Round 1

Reviewer 1 Report

This paper aimed to explore the effects of leaf BSDF effects on LiDAR waveforms, as well as its dependence on wavelength, lidar footprint, view angle, and leaf angle distribution (LAD), by using DIRSIG model. Generally, this work is interesting and meaning. Only several minor revisions should be conducted before publication.

  • I would encourage authors to improve the introduction and the overall description of the technique in such a way to be accessible to a broader audience.
  • There are several parts in section material and methods. Please explain the relationship among various parts.
  • Several Figures should be improved.
  • Other: please check grammatical tense.

Reviewer 2 Report

I would like to congratulate the authors of the article, very interesting and practical paper. In my opinion no changes are needed. Paper ready to publish.

Reviewer 3 Report

This was a very well written paper especially given the topic. The experimental design was clearly defined and the methods of assessment were valid. I found only a few things to change about the paper to add to the clarity, almost all from the standpoint of the figures provided. 

On line 320, there is a ) missing after [74]. 

For the polar coordinate graphs, it would be helpful to provide a clarification of what we should be seeing in these graphs. The figures have an explanation of what components are depicted by what markers, but it would make for easier reading if it was a bit more descriptive. 

Figure 11 there is a discrepancy in the typeface used.

Finally, on all multi-graph figures please reduce the redundancy of text in order to make the graphs easier to read. 
